

# Proteomic indicators of oxidation and hydration state in colorectal cancer

Jeffrey M. Dick

Wattanothaipayap School, Chiang Mai, Thailand

## ABSTRACT

New integrative approaches are needed to harness the potential of rapidly growing datasets of protein expression and microbial community composition in colorectal cancer. Chemical and thermodynamic models offer theoretical tools to describe populations of biomacromolecules and their relative potential for formation in different microenvironmental conditions. The average oxidation state of carbon ($Z_C$) can be calculated as an elemental ratio from the chemical formulas of proteins, and water demand per residue ($\overline{n}_{H_2O}$) is computed by writing the overall formation reactions of proteins from basis species. Using results reported in proteomic studies of clinical samples, many datasets exhibit higher mean $Z_C$ or $\overline{n}_{H_2O}$ of proteins in carcinoma or adenoma compared to normal tissue. In contrast, average protein compositions in bacterial genomes often have lower $Z_C$ for bacteria enriched in fecal samples from cancer patients compared to healthy donors. In thermodynamic calculations, the potential for formation of the cancer-related proteins is energetically favored by changes in the chemical activity of $H_2O$ and fugacity of $O_2$ that reflect the compositional differences. The compositional analysis suggests that a systematic change in chemical composition is an essential feature of cancer proteomes, and the thermodynamic descriptions show that the observed proteomic transformations in host tissue could be promoted by relatively high microenvironmental oxidation and hydration states.

Corresponding author
Jeffrey M. Dick, j3ffdick@gmail.com

## INTRODUCTION

Datasets for differentially expressed proteins in cancer are often interpreted from a mechanistic perspective that emphasizes molecular interactions. Alternative approaches exemplified by recent models that use information theory demonstrate the possibility of interpreting proteomic expression data in a high-level conceptual framework (*Rietman et al.*, *2016*). These approaches may combine concepts from dynamical systems theory and thermodynamics, such as the possible association of "attractor states" in landscape models with low-energy states of a system (*Enver et al.*, *2009*; *Davies, Demetrius & Tuszynski*, *2011*). Despite these advances, energetic functions for differential protein expression have rarely been formulated in terms of physicochemical variables that reflect the conditions of tumor microenvironments. The coupling of recent proteomic data with thermodynamic models using chemical components provides new perspectives on microenvironmental conditions that are conducive to carcinogenesis or healthy growth.

The purpose of the present study is to explore human proteomic and microbial community data for colorectal cancer within a chemical and thermodynamic framework using variables that represent oxidation and hydration state. This is carried out first by comparing chemical compositions of up- and down-expressed proteins along the normal tissue–adenoma–carcinoma progression. Then, a thermodynamic model is used to quantify the overall energetics of the proteomic transformations in terms of chemical potential variables. This approach reveals not only common patterns of chemical changes among many proteomic datasets, but also the possibility that proteomic transformations may be shaped by energetic constraints associated with the changing tumor microenvironment.

Recent years have seen the appearance of many proteomic datasets for colorectal cancer (CRC), a very common and extensively studied type of human cancer. Genomic instability is often considered to be the primary driver of cancer progression (*Kinzler & Vogelstein*, *1996*). However, not only genetic transformations, but also microenvironmental dynamics can influence cancer progression (*Schedin & Elias*, *2004*). Many reactions in the microenvironment, such as those involving hormones or cell–cell signaling interactions, operate on fast timescales, but local hypoxia in tumors and other microenvironmental changes can develop and persist over longer timescales. The long timescales of carcinogenesis may be sufficient for cells to adapt their proteomes to the differential energetic costs of biomolecular synthesis imposed by changing chemical conditions of the microenvironment.

One of the characteristic features of tumors is varying degrees of hypoxia (*Höckel & Vaupel*, *2001*). Hypoxic conditions promote activation of hypoxia-inducible genes by the HIF-1 transcription factor and intensify the mitochondrial generation of reactive oxygen species (ROS) (*Murphy*, *2009*), leading to oxidative stress (*Höckel & Vaupel*, *2001*; *Semenza*, *2008*). It is important to note that there is significant intra-tumor and inter-tumor heterogeneity of oxygenation levels (*Höckel & Vaupel*, *2001*; *DeBerardinis & Cheng*, *2010*). Cancer cells can also exhibit changes in oxidation–reduction (redox) state; for example, redox potential (Eh) monitored *in vivo* in a fibrosarcoma cell line is altered compared to normal fibroblasts (*Hutter, Till & Greene*, *1997*).

The hydration states of cancer cells and tissues may also vary considerably from their healthy counterparts. Microwave detection of differences in dielectric constant resulting from greater water content in malignant tissue is being developed for medical imaging of breast cancer (*Grzegorczyk et al.*, *2012*). IR and Raman spectroscopic techniques also reveal a greater hydration state of cancerous breast tissue, resulting from interaction of water molecules with hydrophilic cellular structures of cancer cells but negligible association with the triglycerides and other hydrophobic molecules that are more common in normal tissue (*Abramczyk et al.*, *2014*).

Increased hydration levels may be associated with a higher abundance of hyaluronan found in the extracellular matrix (ECM) of migrating and metastatic cells (*Toole*, *2002*), while a higher subcellular hydration state may alter cell function by acting as a signal for protein synthesis and cell proliferation (*Häussinger*, *1996*). It has also been hypothesized that the increased hydration of cancer cells underlies a reversion to a more embryonic state (*McIntyre*, *2006*). Based on all of these considerations, compositional and

thermodynamic variables related to redox and hydration state have been selected as the primary descriptive variables in this study.

As noted by others, it seems paradoxical that hypoxia, i.e., low oxygen partial pressure, could be a driving force for the generation of oxidative molecules. Possibly, the mitochondrial generation of ROS is a cellular mechanism for oxygen sensing (*Guzy & Schumacker*, *2006*). Whether through hypoxia-induced oxidative stress or other mechanisms, proteins in cancer have been found to have a variety of oxidative post-translational modifications (PTM), including carbonylation and oxidation of cysteine residues (*Yeh et al.*, *2010*; *Yang et al.*, *2013*). Although proteome-level assessments of oxidative PTM are gaining traction (*Yang et al.*, *2013*), existing large-scale proteomic datasets may carry other signals of oxidation state. One possible "syn-translational" indicator of oxidation state, inherent in the amino acid sequences of proteins, is the average oxidation state of carbon, which is introduced below. At the outset, it is not clear whether such a metric of oxidation state would more closely track hypoxia (i.e., relatively reducing conditions) that may arise in tumors, or a more oxidizing potential connected with ROS and oxidative PTM.

Density functional theory and other computational methods that yield electron density maps of proteins with known structure can be used to compute the partial charges, or oxidation states, of all the atoms. Spectroscopic methods can also be used to determine oxidation states of atoms in molecules (*Gupta et al.*, *2014*). These theoretical and empirical approaches offer the greatest precision in an oxidation state calculation, but it is difficult to apply them to the hundreds of proteins, many with undetermined three-dimensional structures, found to have significantly altered expression in proteomic experiments. Other methods for estimating the oxidation states of atoms in molecules may be needed to assess the overall direction of electron flow in a proteomic transformation.

Some textbooks of organic chemistry present the concept of formal oxidation states, in which the electron pair in a covalent bond is formally assigned to the more electronegative of the two atoms (e.g., *Hendrickson, Cram & Hammond*, *1970*, ch. 18). This rule is consistent with the current IUPAC recommendations for calculating oxidation state of atoms in molecules, but generalizes the IUPAC definitions such that the oxidation states of different carbon atoms in organic molecules can be distinguished (e.g., *Loock*, *2011*; *Gupta et al.*, *2014*). In the primary structure of a protein, where no metal atoms are present and heteroatoms are bonded only to carbon and/or hydrogen, the average oxidation state of carbon ($Z_C$) can be calculated as an elemental ratio, which is easily obtained from the amino acid composition (*Dick*, *2014*). In a protein with the chemical formula $C_c H_h N_n O_o S_s$, the average oxidation state of carbon ($Z_C$) is

$$Z_C = \frac{3n + 2o + 2s - h}{c}.$$ (1)

This equation is equivalent to others, also written in terms of numbers of the elements C, H, N, O and S, used for the average oxidation state of carbon in algal biomass (*Bohutskyi et al.*, *2015*), in humic and fulvic acids (*Fekete et al.*, *2012*), and for the nominal oxidation state of carbon in dissolved organic matter (*Riedel, Biester & Dittmar*, *2012*).

Comparing the average carbon oxidation states in organic molecules is useful for quantifying the reactions of complex mixtures of organic matter in aerosols (*Kroll et al.*, *2011*), the growth of biomass (*Hansen et al.*, *1994*) and the production of biofuels (*Borak, Ort & Burbaum*, *2013*; *Bohutskyi et al.*, *2015*). There is a considerable range of the average oxidation state of carbon in different amino acids (*Masiello et al.*, *2008*; *Amend et al.*, *2013*), with consequences for the energetics of synthesis depending on environmental conditions (*Amend & Shock*, *1998*). Similarly, the nominal oxidation state of carbon can be used as a proxy for the standard Gibbs energies of oxidation reactions of various organic and biochemical molecules (*Arndt et al.*, *2013*). The oxidation state concept can be used as a bookkeeping tool to understand electron flow in metabolic pathways, yet may receive limited coverage in biochemistry courses (*Halkides*, *2000*). There has been scant attention in the literature to the differences in carbon oxidation state among proteins or other biomacromolecules. Nevertheless, the ease of computation makes $Z_C$ a useful metric for rapidly ascertaining the direction and magnitude of electron flow associated with proteomic transformations during disease progression.

Comparisons of oxidation states of carbon can be used to rank the energetics of reactions of organic molecules in particular systems (*Amend et al.*, *2013*). However, quantifying the energetics and mass-balance requirements of chemical transformations requires a more complete thermodynamic model. Thermodynamic models that are based on chemical components (or basis species), i.e., a minimum number of independent chemical formula units that can be combined to form any chemical species in the system, have an established position in geochemistry (*Anderson*, *2005*; *Bethke*, *2008*). The implications of choosing different sets of components, called the "basis" (*Bethke*, *2008*), have received relatively little discussion in biochemistry, although *Alberty* (*2004*) in a similar context highlighted the observation made by *Callen* (*1985*) that "[t]he choice of variables in terms of which a given system is formulated, while seemingly an innocuous step, is often the most crucial step in the solution". Models built with different choices of components nevertheless yield equivalent results when consistently parameterized (*Morel & Hering*, *1993*; *Ravi Kanth et al.*, *2014*). Accordingly, components are a type of chemical accounting for reactions in a system (*Morel & Hering*, *1993*), and do not necessarily constitute mechanistic models for those reactions.

The structure and dynamics of the hydration shells of proteins have important biological consequences (*Levy & Onuchic*, *2006*) and can be investigated in molecular simulation studies (*Wedberg, Abildskov & Peters*, *2012*). Statistical thermodynamics can be used to assess the effects of preferential hydration of protein surfaces on unfolding or other conformational changes (*Lazaridis & Karplus*, *2003*). However, there is also a role for $H_2O$ as a chemical component in stoichiometric reactions representing the mass-balance requirements for formation of proteins with different amino acid sequences.

For example, a system of proteins composed of C, H, N, O and S can be described using the (non-innocuous) components $CO_2$, $NH_3$, $H_2S$, $O_2$ and $H_2O$. Accordingly, stoichiometric reactions representing the formation of certain proteins at the expense of others during a proteomic transformation generally have non-zero coefficients on $O_2$, $H_2O$

and the other components. These stoichiometric reactions can be written without specific knowledge of electron density in proteins or hydration by molecular $H_2O$.

It bears repeating that reactions written using chemical components are not mechanistic representations. Instead, these reactions are specific statements of the requirement for mass balance that can be used to build thermodynamic models of chemically reacting systems (*Helgeson et al.*, *2009*). Flux-balance models of metabolic networks integrate stoichiometric constraints (e.g., *Hiller & Metallo*, *2013*), but stoichiometric descriptions of proteomic transformations are less common, perhaps because of a greater degree of abstraction away from elementary reactions. Nevertheless, the differentially down- and up-expressed proteins in a proteomic dataset furnish a quantitative description of a proteomic transformation and can be viewed as the initial and final states of a chemically reacting system, which is then amenable to thermodynamic modeling.

The chemical potentials of components can be used to describe the internal state of a system and, for an open system, its relation to the environment. Oxygen fugacity is a variable that is related to the chemical potential of $O_2$; it does not necessarily reflect the concentration of $O_2$, but instead indicates the distribution of species with different oxidation states (*Albarède*, *2011*). Theoretical calculation of the energetics of reactions as a function of oxygen fugacity provides a useful reference for the relative stabilities of organic molecules in different environments (*Helgeson et al.*, *2009*; *Amend et al.*, *2013*). However, in a cellular context a multidimensional approach may be required to quantify possible microenvironmental influences on the potentials for biochemical transformations. Likely variables include not only oxidation state but also water activity. Scenarios for early metabolic and cellular evolution (*Pace*, *1991*; *Russell & Hall*, *1997*; *Damer & Deamer*, *2015*) lend additional support to the choice of water activity as a primary variable of interest.

A thermodynamic model that is formulated in terms of carefully selected basis species affords a convenient description of a system. As described in the Methods, a basis is selected that reduces the empirical correlation between average oxidation state of carbon and the coefficient on $H_2O$ in formation reactions of proteins from basis species. The first part of the Results shows compositional comparisons for human and microbial proteins ('Compositional comparisons of human proteins'–'Compositional comparisons of microbial proteins') in 35 datasets from 20 different studies. Many of the comparisons reveal higher $Z_C$ or higher water demand for the formation of proteins up-expressed in cancer compared to normal tissue. Contrary to the trends observed for human proteins, the average protein compositions of bacteria enriched in cancer tend to have lower $Z_C$.

To better understand the biochemical context of these differences, calculations reported in the second part of the Results use chemical affinity (negative Gibbs energy of reaction) to predict the most stable molecules as a function of oxygen fugacity and water activity ('Thermodynamic descriptions: background'–'Relative stability fields for human proteins'). Theoretical calculations of the relative stabilities of groups of up- and down-expressed proteins build on the compositional descriptions as a step toward quantifying the microenvironmental conditions that may promote or impede the proteomic alterations associated with the progression of cancer.

## METHODS

### Data sources

This section describes the data sources and additional data processing steps applied in this study. An attempt was made to locate all currently available proteomic studies for clinical tissue on CRC including, among others, those listed in the "Tissue" and "Tissue subproteomes" sections of the review paper by *De Wit et al.* (*2013*) and in Supporting Table 3 ("Clinical Samples") of the review paper by *Martínez-Aguilar et al.* (*2013*). To make the comparisons more robust, only datasets with at least 30 proteins in each of the up- and down-regulated groups were considered; however, all datasets from a given study were included if at least one of the datasets met this criterion. The reference keys for the selected studies shown below and in Table 1 are derived from the names of the authors and year of publication.

In comparisons between groups of up- and down-expressed proteins, the convention in this study is to consider proteins with higher expression in normal tissue or less-advanced cancer stages as a "normal" group (group 1), with number of proteins $n_1$, while proteins with higher expression in cancer or more-advanced cancer stages are categorized as a "cancer" group (group 2), with number of proteins $n_2$. For example, in the dataset of *Uzozie et al.* (*2014*) comparing normal mucosa and adenoma, the proteins up-expressed in adenoma are assigned to group 2, while in the adenoma—carcinoma dataset of *Knol et al.* (*2014*), the proteins with higher expression in adenoma are assigned to group 1 (see Table 1).

Names or IDs of genes or proteins given in the sources were searched in UniProt (*The UniProt Consortium*, *2015*). The corresponding UniProt IDs are provided in the `*.csv` data files in Dataset S1. Amino acid sequences of human proteins were taken from the UniProt reference proteome (files `UP000005640_9606.fasta.gz` containing canonical, manually reviewed sequences, and `UP000005640_9606_additional.fasta.gz` containing isoforms and unreviewed sequences, dated 2016-04-13, downloaded from ftp://ftp.uniprot.org/pub/databases/uniprot/current_release/knowledgebase/reference_proteomes/Eukaryota/). Entire sequences were used; i.e., signal peptides and propeptides were not removed when calculating the amino acid compositions. However, amino acid compositions were calculated for particular isoforms, if these were identified in the sources. Files `human.aa.csv` and `human_additional.aa.csv` in Dataset S1 contain the amino acid compositions of the proteins calculated from the UniProt reference proteome. In a few cases, amino acid compositions of unreviewed or obsolete sequences in UniProt, not available in the reference proteome, were individually compiled; these are contained in file `human2.aa.csv` in Dataset S1.

Reported gene names were converted to UniProt IDs using the UniProt mapping tool (http://www.uniprot.org/mapping), and IPI accession numbers were converted to UniProt IDs using the DAVID conversion tool (https://david.ncifcrf.gov/content.jsp?file=conversion.html). For proteins with no automatically generated matches, manual searches in UniProt of the protein descriptions, where available, were performed. Proteins with missing or duplicated identifiers, or those that could not be matched to a UniProt ID, were omitted from the comparisons here. Therefore, the numbers of proteins actually used in

![PeerJ]

**Table 1  Summary of compositional comparisons for human proteins.** Mean differences (MD), percent values of common language effect size (ES), and $p$-values are shown for comparisons between groups of proteins reported to have higher abundance in normal ($n_1$) or cancer ($n_2$) tissue (or less advanced or more advanced cancer stages, respectively). The textual descriptions are written such that the ordering around the slash ("/") corresponds to $n_2/n_1$. References and specific abbreviations used in the descriptions are given in 'Data sources'.

| Reference (Description) | $n_1$ | $n_2$ | $Z_C$ | | | $\bar{n}_{H_2O}$ | | |
|---|---|---|---|---|---|---|---|---|
| | | | MD | ES | $p$-value | MD | ES | $p$-value |
| WTK+08 (T/N) | 57 | 70 | 0.018 | 55 | 3e–01 | 0.006 | 52 | 7e–01 |
| AKP+10 (CRC nuclear matrix C/A) | 101 | 28 | −0.012 | 47 | 7e–01 | −0.009 | 48 | 8e–01 |
| AKP+10 (CIN nuclear matrix C/A) | 87 | 81 | **−0.031** | **40** | **3e–02** | 0.006 | 48 | 7e–01 |
| AKP+10 (MIN nuclear matrix C/A) | 157 | 76 | −0.002 | 52 | 7e–01 | −0.013 | 45 | 3e–01 |
| JKMF10 (serum biomarkers up/down) | 43 | 56 | −0.007 | 46 | 5e–01 | **0.056** | **67** | **4e–03** |
| XZC+10 (stage I/normal) | 48 | 166 | 0.009 | 52 | 7e–01 | 0.026 | 56 | 2e–01 |
| XZC+10 (stage II/normal) | 77 | 321 | **0.022** | **60** | **6e–03** | 0.018 | 54 | 3e–01 |
| ZYS+10 (microdissected T/N) | 60 | 57 | 0.019 | 58 | 1e–01 | 0.022 | 58 | 1e–01 |
| BPV+11 (adenoma/normal) | 71 | 92 | **−0.023** | **40** | **4e–02** | 0.004 | 49 | 8e–01 |
| BPV+11 (stage I/normal) | 109 | 72 | -0.007 | 47 | 5e–01 | 0.005 | 50 | 9e–01 |
| BPV+11 (stage II/normal) | 164 | 140 | **0.031** | **62** | **3e–04** | 0.006 | 51 | 7e–01 |
| BPV+11 (stage III/normal) | 63 | 131 | **0.025** | **62** | **9e–03** | −0.005 | 47 | 5e–01 |
| BPV+11 (stage IV/normal) | 42 | 26 | −0.010 | 44 | 4e–01 | 0.005 | 52 | 8e–01 |
| JCF+11 (T/N) | 72 | 45 | **0.032** | **63** | **2e–02** | −0.003 | 49 | 8e–01 |
| MRK+11 (adenoma/normal) | 335 | 288 | 0.011 | 54 | 1e–01 | **0.058** | **68** | **2e–15** |
| MRK+11 (adenocarcinoma/adenoma) | 373 | 257 | **0.034** | **65** | **1e–10** | −0.009 | 47 | 1e–01 |
| MRK+11 (adenocarcinoma/normal) | 351 | 232 | **0.034** | **63** | **4e–08** | **0.035** | **61** | **8e–06** |
| KKL+12 (poor/good prognosis) | 75 | 61 | **0.026** | **64** | **5e–03** | −0.002 | 48 | 8e–01 |
| KYK+12 (MSS-type T/N) | 73 | 175 | **0.024** | **61** | **9e–03** | 0.023 | 56 | 1e–01 |
| WOD+12 (T/N) | 79 | 677 | 0.016 | 54 | 2e–01 | 0.027 | 58 | **2e–02** |
| YLZ+12 (conditioned media T/N) | 55 | 68 | **0.024** | **61** | **4e–02** | 0.009 | 54 | 5e–01 |
| MCZ+13 (stromal T/N) | 33 | 37 | **0.047** | **74** | **5e–04** | −0.034 | 42 | 2e–01 |
| KWA+14 (chromatin-binding C/A) | 51 | 55 | **−0.039** | **29** | **2e–04** | −0.010 | 48 | 7e–01 |
| UNS+14 (epithelial adenoma/normal) | 58 | 65 | 0.001 | 49 | 8e–01 | **0.032** | **61** | **4e–02** |
| WKP+14 (tissue secretome T/N) | 44 | 210 | 0.006 | 53 | 6e–01 | **0.057** | **68** | **1e–04** |
| STK+15 (membrane enriched T/N) | 113 | 66 | 0.005 | 52 | 6e–01 | 0.025 | 55 | 2e–01 |
| WDO+15 (adenoma/normal) | 1,061 | 1,254 | **0.030** | **64** | **7e–33** | 0.023 | 58 | **7e–11** |
| WDO+15 (carcinoma/adenoma) | 772 | 1,007 | −0.013 | 42 | **2e–08** | −0.003 | 50 | 7e–01 |
| WDO+15 (carcinoma/normal) | 879 | 1,281 | 0.014 | 57 | **9e–08** | 0.024 | 58 | **1e–10** |
| LPL+16 (stromal AD/NC) | 123 | 75 | **−0.039** | **32** | **2e–05** | **0.037** | **60** | **2e–02** |
| LPL+16 (stromal CIS/NC) | 125 | 60 | −0.007 | 46 | 4e–01 | −0.001 | 52 | 7e–01 |
| LPL+16 (stromal ICC/NC) | 99 | 75 | 0.001 | 47 | 6e–01 | −0.021 | 48 | 7e–01 |
| PHL+16 (AD/NC) | 113 | 86 | 0.011 | 54 | 4e–01 | **0.037** | **60** | **2e–02** |
| PHL+16 (CIS/NC) | 169 | 138 | 0.019 | 59 | **5e–03** | 0.001 | 49 | 7e–01 |
| PHL+16 (ICC/NC) | 129 | 100 | 0.016 | 57 | 6e–02 | −0.007 | 46 | 3e–01 |

**Notes.**

Abbreviations: T/N, tumor/normal; C/A, carcinoma/adenoma.

the comparisons (listed in Table 1) may be different from the numbers of proteins reported by the authors and summarized below.

WTK+08: *Watanabe et al.* (*2008*) used 2-nitrobenzenesulfenyl labeling and MS/MS analysis to identify 128 proteins with differential expression in paired CRC and normal tissue specimens from 12 patients. The list of proteins used in this study was generated by combining the lists of up- and down-regulated proteins from Table 1 and Supplementary Data 1 of *Watanabe et al.* (*2008*) with the Swiss-Prot and UniProt accession numbers from their Supplementary Data 2.

AKP+10: *Albrethsen et al.* (*2010*) used nano-LC-MS/MS to characterize proteins from the nuclear matrix fraction in samples from 2 patients each with adenoma (ADE), chromosomal instability CRC (CIN+) and microsatellite instability CRC (MIN+). Cluster analysis was used to classify proteins with differential expression between ADE and CIN+, MIN+, or in both subtypes of carcinoma (CRC). Here, gene names from Supplementary Tables 5–7 of *Albrethsen et al.* (*2010*) were converted to UniProt IDs using the UniProt mapping tool.

JKMF10: *Jimenez et al.* (*2010*) compiled a list of candidate serum biomarkers from a meta-analysis of the literature. In the meta-analysis, 99 up- or down-expressed proteins were identified in at least 2 studies. The list of UniProt IDs used in this study was taken from Table 4 of *Jimenez et al.* (*2010*).

XZC+10: *Xie et al.* (*2010*) used a gel-enhanced LC-MS method to analyze proteins in pooled tissue samples from 13 stage I and 24 stage II CRC patients and pooled normal colonic tissues from the same patients. Here, IPI accession numbers from Supplemental Table 4 of *Xie et al.* (*2010*) were converted to UniProt IDs using the DAVID conversion tool.

ZYS+10: *Zhang et al.* (*2010*) used acetylation stable isotope labeling and LTQ-FT MS to analyze proteins in pooled microdissected epithelial samples of tumor and normal mucosa from 20 patients, finding 67 and 70 proteins with increased or decreased expression (ratios $\geq 2$ or $\leq 0.5$). Here, IPI accession numbers from Supplemental Table 4 of *Zhang et al.* (*2010*) were converted to UniProt IDs using the DAVID conversion tool.

BPV+11: *Besson et al.* (*2011*) analyzed microdissected cancer and normal tissues from 28 patients (4 adenoma samples and 24 CRC samples at different stages) using iTRAQ labeling and MALDI-TOF/TOF MS to identify 555 proteins with differential expression between adenoma and stage I, II, III, IV CRC. Here, gene names from supplemental Table 9 of *Besson et al.* (*2011*) were converted to UniProt IDs using the UniProt mapping tool.

JCF+11: *Jankova et al.* (*2011*) analyzed paired samples from 16 patients using iTRAQ-MS to identify 118 proteins with >1.3-fold differential expression between CRC tumors and adjacent normal mucosa. The protein list used in this study was taken from Supplementary Table 2 of *Jankova et al.* (*2011*).

MRK+11: *Mikula et al.* (*2011*) used iTRAQ labeling with LC-MS/MS to identify a total of 1,061 proteins with differential expression (fold change $\geq 1.5$ and false discovery rate $\leq 0.01$) between pooled samples of 4 normal colon (NC), 12 tubular or tubulo-villous adenoma (AD) and 5 adenocarcinoma (AC) tissues. The list of proteins used in this study was taken from Table S8 of *Mikula et al.* (*2011*).

KKL+12: *Kim et al.* (*2012*) used difference in-gel electrophoresis (DIGE) and cleavable isotope-coded affinity tag (cICAT) labeling followed by mass spectrometry to identify

175 proteins with more than 2-fold abundance ratios between microdissected and pooled tumor tissues from stage-IV CRC patients with good outcomes (survived more than five years; 3 patients) and poor outcomes (died within 25 months; 3 patients). The protein list used in this study was made by filtering the cICAT data from Supplementary Table 5 of *Kim et al.* (*2012*) with an abundance ratio cutoff of >2 or <0.5, giving 147 proteins. IPI accession numbers were converted to UniProt IDs using the DAVID conversion tool.

KYK+12: *Kang et al.* (*2012*) used mTRAQ and cICAT analysis of pooled microsatellite stable (MSS-type) CRC tissues and pooled matched normal tissues from 3 patients to identify 1,009 and 478 proteins in cancer tissue with increased or decreased expression by higher than 2-fold, respectively. Here, the list of proteins from Supplementary Table 4 of *Kang et al.* (*2012*) was filtered to include proteins with expression ratio >2 or <0.5 in both mTRAQ and cICAT analyses, leaving 175 up-expressed and 248 down-expressed proteins in CRC. Gene names were converted to UniProt IDs using the UniProt mapping tool.

WOD+12: *Wiśniewski et al.* (*2012*) used LC-MS/MS to analyze proteins in microdissected samples of formalin-fixed paraffin-embedded (FFPE) tissue from 8 patients; at $P < 0.01$, 762 proteins had differential expression between normal mucosa and primary tumors. The list of proteins used in this study was taken from Supplementary Table 4 of *Wiśniewski et al.* (*2012*).

YLZ+12: *Yao et al.* (*2012*) analyzed the conditioned media of paired stage I or IIA CRC and normal tissues from 9 patients using lectin affinity capture for glycoprotein (secreted protein) enrichment by nano LC-MS/MS to identify 68 up-regulated and 55 down-regulated differentially expressed proteins. IPI accession numbers listed in Supplementary Table 2 of *Yao et al.* (*2012*) were converted to UniProt IDs using the DAVID conversion tool.

MCZ+13: *Mu et al.* (*2013*) used laser capture microdissection (LCM) to separate stromal cells from 8 colon adenocarcinoma and 8 non-neoplastic tissue samples, which were pooled and analyzed by iTRAQ to identify 70 differentially expressed proteins. Here, gi numbers listed in Table 1 of *Mu et al.* (*2013*) were converted to UniProt IDs using the UniProt mapping tool; FASTA sequences of 31 proteins not found in UniProt were downloaded from NCBI and amino acid compositions were added to `human2.aa.csv`.

KWA+14: *Knol et al.* (*2014*) used differential biochemical extraction to isolate the chromatin-binding fraction in frozen samples of colon adenomas (3 patients) and carcinomas (5 patients), and LC-MS/MS was used for protein identification and label-free quantification. The results were combined with a database search to generate a list of 106 proteins with nuclear annotations and at least a three-fold expression difference. Here, gene names from Table 2 of *Knol et al.* (*2014*) were converted to UniProt IDs.

UNS+14: *Uzozie et al.* (*2014*) analyzed 30 samples of colorectal adenomas and paired normal mucosa using iTRAQ labeling, OFFGEL electrophoresis and LC-MS/MS. 111 proteins with expression fold changes ($\log_2$) at least ±0.5 and statistical significance threshold $q < 0.02$ that were also quantified in cell-line experiments were classified as "epithelial cell signature proteins". UniProt IDs were taken from Table III of *Uzozie et al.* (*2014*).

WKP+14: *de Wit et al.* (*2014*) analyzed the secretome of paired CRC and normal tissue from 4 patients, adopting a five-fold enrichment cutoff for identification of candidate biomarkers. Here, the list of proteins from Supplementary Table 1 of *de Wit et al.* (*2014*)

was filtered to include those with at least five-fold greater or lower abundance in CRC samples and $p < 0.05$. Two proteins listed as "Unmapped by Ingenuity" were removed, and gene names were converted to UniProt IDs using the UniProt mapping tool.

STK+15: *Sethi et al.* (*2015*) analyzed the membrane-enriched proteome from tumor and adjacent normal tissues from 8 patients using label-free nano-LC-MS/MS to identify 184 proteins with a fold change > 1.5 and *p*-value < 0.05. Here, protein identifiers from Supporting Table 2 of *Sethi et al.* (*2015*) were used to find the corresponding UniProt IDs.

WDO+15: *Wiśniewski et al.* (*2015*) analyzed 8 matched formalin-fixed and paraffin-embedded (FFPE) samples of normal tissue (N) and adenocarcinoma (C) and 16 nonmatched adenoma samples (A) using LC-MS to identify 2300 (N/A), 1780 (A/C) and 2161 (N/C) up- or down-regulated proteins at $p < 0.05$. The list of proteins used in this study includes only those marked as having a significant change in SI Table 3 of *Wiśniewski et al.* (*2015*).

LPL+16: *Li et al.* (*2016*) used iTRAQ and 2D LC-MS/MS to analyze pooled samples of stroma purified by laser capture microdissection (LCM) from 5 cases of non-neoplastic colonic mucosa (NC), 8 of adenomatous colon polyps (AD), 5 of colon carcinoma *in situ* (CIS) and 9 of invasive colonic carcinoma (ICC). A total of 222 differentially expressed proteins between NC and other stages were identified. Here, gene symbols from Supplementary Table S3 of *Li et al.* (*2016*) were converted to UniProt IDs using the UniProt mapping tool.

PHL+16: *Peng et al.* (*2016*) used iTRAQ 2D LC-MS/MS to analyze pooled samples from 5 cases of normal colonic mucosa (NC), 8 of adenoma (AD), 5 of carcinoma *in situ* (CIS) and 9 of invasive colorectal cancer (ICC). A total of 326 proteins with differential expression between two successive stages (and, for CIS and ICC, also differentially expressed with respect to NC) were detected. The list of proteins used in this study was generated by converting the gene names in Supplementary Table 4 of *Peng et al.* (*2016*) to UniProt IDs using the UniProt mapping tool.

## Basis I

To formulate a thermodynamic description of a chemically reacting system, an important choice must be made regarding the basis species used to describe the system. The basis species, like thermodynamic components, are a minimum number of chemical formula units that can be linearly combined to generate the composition of any chemical species in the system of interest. Stated differently, any species can be formed by combining the components, but components can not be used to form other components (*VanBriesen & Rittmann*, *1999*). Within these constraints, any specific choice of a basis is theoretically permissible. In making the choice of components, convenience (*Gibbs*, *1875*), ease of interpretation and relationship with measurable variables, as well as availability of thermodynamic data (e.g., *Helgeson*, *1970*), and kinetic favorability (*May & Murray*, *2001*) are other useful considerations. Once the basis species are chosen, the stoichiometric coefficients in the formation reaction for any chemical species are algebraically determined.

Following previous studies (e.g., *Dick*, *2008*), the basis species initially chosen here are $CO_2$, $H_2O$, $NH_3$, $H_2S$ and $O_2$ (Basis I). The reaction representing the overall formation

from these basis species of a protein having the formula $C_cH_hN_nO_oS_s$ is

$$cCO_2 + nNH_3 + sH_2S + n_{H_2O}H_2O + n_{O_2}O_2 \rightarrow C_cH_hN_nO_oS_s \qquad \text{(R1)}$$

where $n_{H_2O} = (h - 3n - 2s)/2$ and $n_{O_2} = (o - 2c - n_{H_2O})/2$. Dividing $n_{H_2O}$ by the length of the protein gives the water demand per residue ($\overline{n}_{H_2O}$), which is used here because proteins in the comparisons generally have different sequence lengths.

These or similar sets of inorganic species (such as $H_2$ instead of $O_2$) are often used in studying reaction energetics in geobiochemistry (e.g., *Shock & Canovas*, *2010*). However, as seen in Figs. 1A and 1B, there is a high correlation between $Z_C$ of protein molecules and $\overline{n}_{H_2O}$ in the reactions to form the proteins from Basis I (note that the choice of basis species affects only $\overline{n}_{H_2O}$ and not $Z_C$). Because of this stoichiometric interdependence, changes in either redox or hydration potential, while holding the chemical potentials of the remaining basis species constant, have correlated effects on the energetics of chemical transformations (see 'Comparison with inorganic basis species' below). A different set of basis species can be chosen that reduces this correlation and affords a more informative description of the compositional changes in proteomic transformations.

## Basis II

In this exploratory study, we restrict attention to at most two variables, with the implication that the others are held constant. In a subcellular setting, assuming that the chemical potentials of $CO_2$, $NH_3$ and $H_2S$ do not change during a proteomic transformation, as implied by varying the chemical potentials of $O_2$ and $H_2O$ in Basis I, may be less appropriate than assuming constant chemical potentials of more complex metabolites. In thermodynamic models for systems of proteins, constant chemical activities of chemical components having the compositions of amino acids might be a reasonable provision.

Although 1140 3-way combinations can be made of the 20 common proteinogenic amino acids, only 324 of the combinations contain cysteine and/or methionine (one of these is required to provide sulfur), and of these only 300, when combined with $O_2$ and $H_2O$, are compositionally nondegenerate. The slope, intercept and $R^2$ of the linear least-squares fits between $Z_C$ and $\overline{n}_{H_2O}$ using each possible basis containing $O_2$, $H_2O$ and three amino acids are listed in file AAbasis.csv in Dataset S1. Many of these combinations have lower $R^2$ and lower slopes than found for Basis I (Figs. 1A and 1B), indicating a decreased correlation. From those with a lower correlation, but not the lowest, the basis including cysteine (Cys), glutamic acid (Glu), glutamine (Gln), $O_2$ and $H_2O$ (Basis II) has been selected for use in this study. The scatterplots and fits between $Z_C$ and $\overline{n}_{H_2O}$ using Basis II are shown in Figs. 1C and 1D.

A secondary consideration in choosing this basis instead of others with even lower $R^2$ is the centrality of glutamine and glutamic acid in many metabolic pathways (e.g., *DeBerardinis & Cheng*, *2010*). Accordingly, these amino acids may be kinetically more reactive than others in pathways of protein synthesis and degradation. The presence of side chains derived from cysteine and glutamic acid in the abundant glutathione molecule (GSH), associated with redox homeostasis, is also suggestive of a central metabolic requirement for these amino acids. Again, it must be stressed that the current provisional choice of

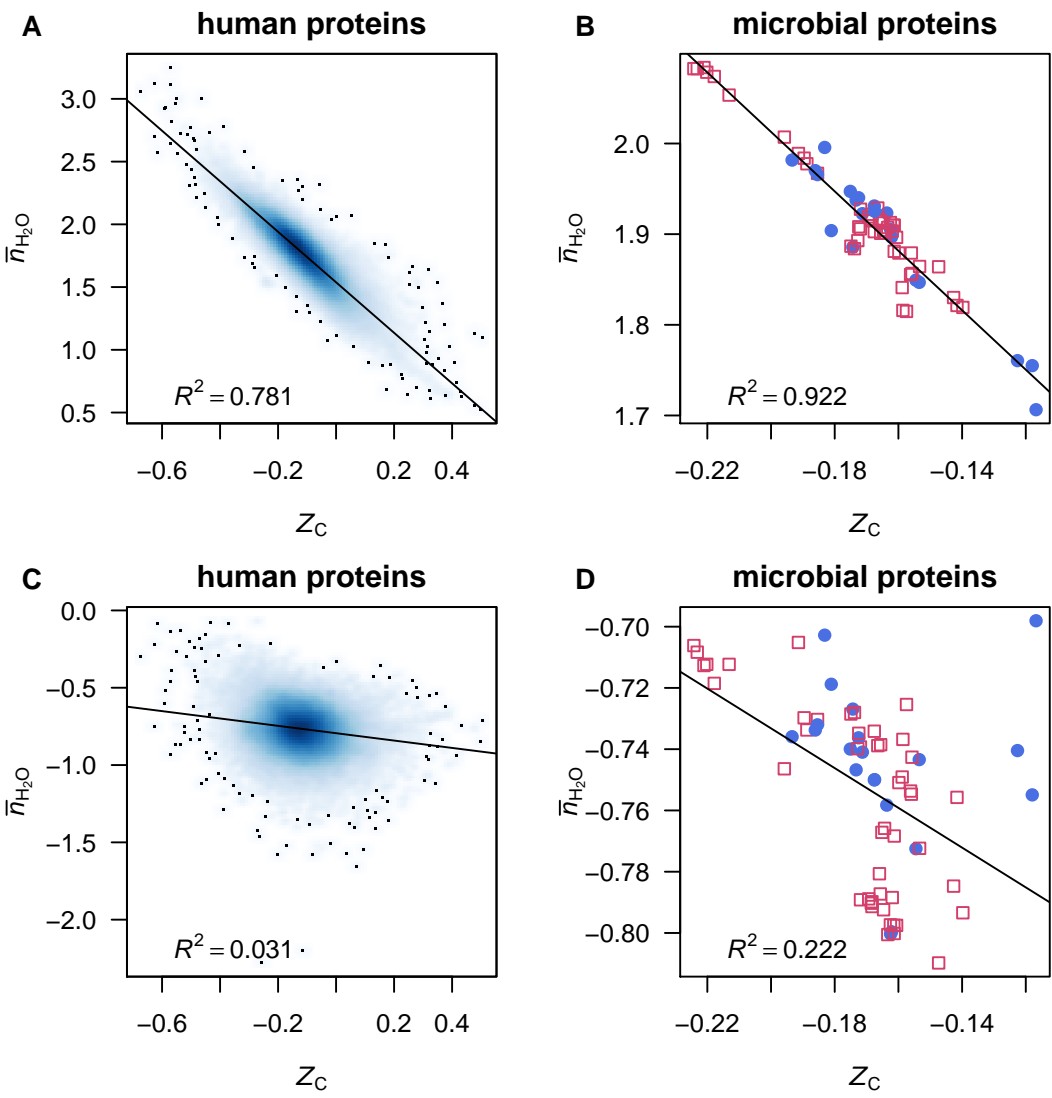

**Figure 1** Scatterplots of average oxidation state of carbon ($Z_C$) and water demand per residue ($\overline{n}_{H_2O}$). Data are plotted for (A, C) individual human proteins and (B,D) mean composition of proteins from microbial genomes, with $\overline{n}_{H_2O}$ computed using (A, B) Basis I (Reaction (R1)) or (C, D) Basis II (Reaction (R2)). Linear least-squares fits and $R^2$ values are shown. In (A) and (C), the intensity of shading corresponds to density of points, produced using the smoothScatter() function of R graphics (*R Core Team*, *2016*). The label in plot (A) identifies a particular protein, MUC1, which is used for the example calculations (see Reactions (R3) and (R4)).

basis species is neither uniquely determined nor necessarily optimal for a thermodynamic description of any particular system. More experience with thermodynamic modeling and better biochemical intuition will likely provide reasons to refine these calculations using a different basis, perhaps including metabolites other than amino acids.

A general formation reaction using Basis II is

$$n_{Cys}C_3H_7NO_2S + n_{Glu}C_5H_9NO_4 + n_{Gln}C_5H_{10}N_2O_3$$
$$+ n_{H_2O}H_2O + n_{O_2}O_2 \rightarrow C_cH_hN_nO_oS_s \tag{R2}$$

where the reaction coefficients ($n_{Cys}$, $n_{Glu}$, $n_{Gln}$, $n_{H_2O}$ and $n_{O_2}$) can be obtained by solving

$$\begin{bmatrix} 3 & 5 & 5 & 0 & 0 \\ 7 & 9 & 10 & 2 & 0 \\ 1 & 1 & 2 & 0 & 0 \\ 2 & 4 & 3 & 1 & 2 \\ 1 & 0 & 0 & 0 & 0 \end{bmatrix} \times \begin{bmatrix} n_{Cys} \\ n_{Glu} \\ n_{Gln} \\ n_{H_2O} \\ n_{O_2} \end{bmatrix} = \begin{bmatrix} c \\ h \\ n \\ o \\ s \end{bmatrix}. \tag{2}$$

Although the definition of basis species requires that they are themselves compositionally nondegenerate, the matrix equation emphasizes the interdependence of the stoichiometric reaction coefficients. A consequence of this multiple dependence is that single variables such as $n_{O_2}$ and $n_{H_2O}$ are not simple variables, but are influenced by both the chemical composition of the protein and the choice of basis species used to describe the system.

The combination of molecules shown in Reaction (R2) does not represent the actual mechanism of synthesis of the proteins. Instead, reactions such as this account for mass-conservation requirements and permit the subsequent generation of thermodynamic models for the potential for formation of different proteins as a function of system parameters (i.e., chemical potentials of $O_2$ and $H_2O$).

As an example of a specific calculation, consider the following reaction for MUC1, a chromatin-binding protein that is highly up-expressed in CRC cells (*Knol et al.*, *2014*).

$$7C_3H_7NO_2S + 535.6C_5H_9NO_4 + 515.2C_5H_{10}N_2O_3$$
$$\rightarrow C_{5275}H_{8231}N_{1573}O_{1762}S_7 + 895.2H_2O + 522.4O_2. \tag{R3}$$

As with the other reactions shown above, this reaction is not a mechanism, but represents the stoichiometric requirements for the formation from the basis species of one mole of the protein. Water is released in Reaction (R3), so the water demand ($n_{H_2O}$) is negative. The length of this protein is 1,255 amino acid residues, giving the water demand per residue, $\overline{n}_{H_2O} = -895.2/1,255 = -0.71$. The average oxidation state of carbon ($Z_C$) in MUC1, which does not depend on the choice of basis species, is 0.005 (Eq. 1). The value of $Z_C$ indicates that MUC1 is a relatively highly oxidized protein, while its $\overline{n}_{H_2O}$ places it near the median water demand for up-expressed proteins in cancer in this dataset (see Fig. 2A below).

### Thermodynamic calculations

Standard molal thermodynamic properties of the amino acids and unfolded proteins estimated using amino acid group additivity were calculated as described by *Dick, LaRowe & Helgeson* (*2006*), taking account of updated values for the methionine sidechain group (*LaRowe & Dick*, *2012*). In this study, the Gibbs energies of hypothetically non-ionized proteins were used, and calculations were carried out at 37 °C and 1 bar. The temperature dependence of standard Gibbs energies was calculated using the revised Helgeson–Kirkham–Flowers (HKF) equations of state (*Helgeson, Kirkham & Flowers*, *1981*; *Tanger IV & Helgeson*, *1988*). Thermodynamic properties for $O_2$ (gas) were calculated using data from *Wagman et al.* (*1982*) and the Maier–Kelley heat capacity function (*Kelley*, *1960*). Properties of liquid $H_2O$ were calculated using data and extrapolations coded in Fortran

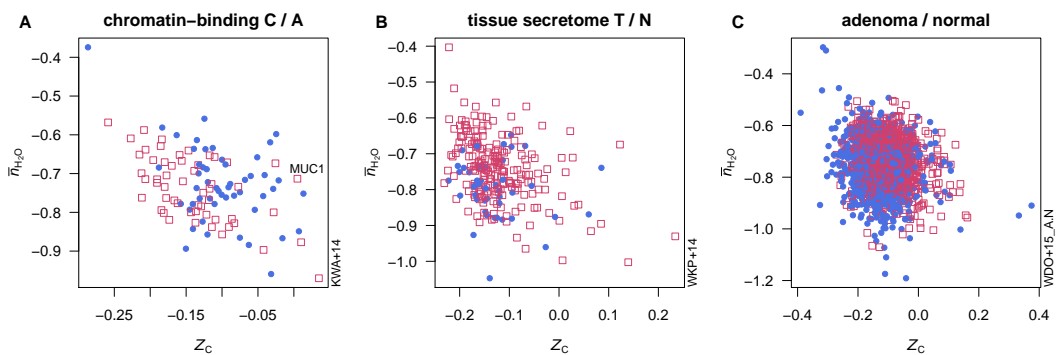

**Figure 2** **Average oxidation state of carbon ($Z_C$) and water demand per residue ($\overline{n}_{H_2O}$) for proteins in selected datasets.** Open red squares represent proteins enriched in tumors or more advanced cancer stages, and filled blue circles represent proteins enriched in normal tissue or less advanced cancer stages.

subroutines from the SUPCRT92 package (*Johnson, Oelkers & Helgeson*, *1992*), as provided in the CHNOSZ package (*Dick*, *2008*).

Chemical affinities of reactions were calculated using activities of amino acids in the basis equal to $10^{-4}$, and activities of proteins equal to 1/(protein length) (i.e., unit activity of amino acid residues). Continuing with the example of Reaction (R3), an estimate of the standard Gibbs energy ($\Delta G_f^o$) of the aqueous protein molecule (*Dick, LaRowe & Helgeson*, *2006*; *LaRowe & Dick*, *2012*) at 37 °C is $-40,974$ kcal/mol; combined with the standard Gibbs energies of the basis species, this give a standard Gibbs energy of reaction ($\Delta G_r^o$) equal to 66,889 kcal/mol. At $\log a_{H_2O} = 0$ and $\log f_{O_2} = -65$, with activities of the amino acid basis species equal to $10^{-4}$, the overall Gibbs energy ($\Delta G_r$) is 24,701 kcal/mol. The negative of this value is the chemical affinity ($A$) of the reaction. The per-residue chemical affinity for formation of protein MUC1 in the stated conditions is $-19.7$ kcal/mol. (This calculation can be reproduced using the function `reaction()` in file `plot.R` in Dataset S1.)

In a given system, proteins with higher (more positive) chemical affinity are relatively energetically stabilized, and theoretically have a higher propensity to be formed. Therefore, the differences in affinities reflect not only the amino acid compositions of the protein molecules but also the potential for local environmental conditions to influence the relative abundances of proteins.

### Weighted rank difference

The contours on relative stability diagrams for the groups of differentially expressed proteins (see Fig. 6 below) depict the weighted rank differences of chemical affinities of formation of proteins. To illustrate this calculation, consider a hypothetical system composed of 3 proteins with higher expression in cancer (C) and 4 with higher expression in normal samples (down-expressed in cancer, i.e., having higher expression in a healthy state) (H). Suppose that under one set of conditions (i.e., specified $\log a_{H_2O}$ and $\log f_{O_2}$), the per-residue affinities of the proteins give the following ranking in ascending order (I):

C  C  C  H  H  H  H
1  2  3  4  5  6  7

This gives as the sum of ranks for up-expressed (C) proteins $\sum r_C = 6$, and for down-expressed (H) proteins $\sum r_H = 22$. The difference in sum of ranks is $\Delta r_{C-H} = -16$; the negative value is associated with a higher rank sum for the down-expressed proteins, indicating that these as a group are more stable than the down-expressed proteins. In a second set of conditions, we might have (II):

H   H   H   H   C   C   C
1   2   3   4   5   6   7

Here, the difference of rank sums is $\Delta r_{C-H} = 18 - 10 = 8$.

For systems where the numbers of proteins in the two groups are equal, the maximum possible differences in rank sums would have equal absolute values, but that is not the case in this and other systems having unequal numbers of up- and down-expressed proteins. To characterize these datasets, the weighted rank-sum difference can be calculated using

$$\Delta \bar{r} = 2\left( \frac{n_H}{n} \sum r_C - \frac{n_C}{n} \sum r_H \right) \tag{3}$$

where $n_H$, $n_C$ and $n$ are the numbers of down-expressed, up-expressed, and total proteins in the comparison. In the example here, we have $n_H/n = 4/7$ and $n_C/n = 3/7$. Equation (3) then gives $\Delta \bar{r} = -12$ and $\Delta \bar{r} = 12$, respectively, for conditions (I) and (II) above, showing equal weighted rank-sum differences for the two extreme rankings.

We can also consider a situation where the ranks of the proteins are evenly distributed:

H   C   H   C   H   C   H
1   2   3   4   5   6   7

Here the absolute difference of rank sums is $\Delta r_{C-H} = 12 - 16 = -4$, but the weighted rank-sum difference is $\Delta \bar{r} = 0$. The zero value for an even distribution and the opposite values for the two extremes demonstrate the applicability of this weighting scheme.

### Software and data availability

All statistical and thermodynamic calculations were performed using R (*R Core Team*, *2016*). Thermodynamic calculations were carried out using R package CHNOSZ (*Dick*, *2008*). Effect sizes (see below) were calculated using R package orddom (*Rogmann*, *2013*). Figures were generated using CHNOSZ and graphical functions available in R together with the R package colorspace (*Ihaka et al.*, *2015*) for constructing an HCL-based color palette (*Zeileis, Hornik & Murrell*, *2009*). With the mentioned packages installed, Table 1 and the figures in this paper can be reproduced using the code (`plot.R`) and data files (`*.csv`) in Dataset S1.

## RESULTS

### Compositional comparisons of human proteins

Comparisons of proteome composition in terms of average oxidation state of carbon ($Z_C$) and water demand per residue ($\bar{n}_{H_2O}$) are presented in Fig. 2 and Table 1. Figure 2 shows scatterplots of individual protein compositions for proteomes in three representative

studies. Each of these exhibits a strongly differential trend in $Z_C$ or $\bar{n}_{H_2O}$ that can be visually identified. In Fig. 2A, chromatin-binding proteins highly expressed in carcinoma (*Knol et al.*, *2014*) as a group exhibit a lower $Z_C$ than those found to be more abundant in adenoma. In Fig. 2B, proteins relatively highly expressed in epithelial cells in adenoma (*Uzozie et al.*, *2014*) tend to have higher $\bar{n}_{H_2O}$ than the proteins more highly expressed in paired normal tissues. Differentially expressed proteins between adenoma and normal tissue identified in a recent deep-proteome analysis (*Wiśniewski et al.*, *2015*) are compared in Fig. 2C, showing that proteins up-expressed in adenoma are relatively oxidized (i.e., have higher $Z_C$).

In order to quantify these differences, Table 1 shows the numbers of proteins in each comparison ($n_1$ for normal or less advanced cancer stage; $n_2$ for tumor or more advanced cancer stage), differences of means (MD), common language effect size as percentages (ES), and $p$-values calculated using the Wilcoxon rank sum test. This non-parametric test is suitable for data which may not be normally distributed. For a given experiment, the common language effect size, or probability of superiority, describes the probability that $Z_C$ or $\bar{n}_{H_2O}$ of a protein is higher in the cancer group than in the normal group. That is, percent values of the ES greater than (or less than) 50 indicate a greater proportion of pairwise higher (or lower) $Z_C$ or $\bar{n}_{H_2O}$ of proteins in the $n_2$ compared to $n_1$ groups. The ES and $p$-value are used here to allow for a subjective assessment of the compositional differences. ES values $\geq 60$ or $\leq 40$ and $p$-values $< 0.05$ are highlighted in the table. The corresponding mean differences are underlined for $p < 0.05$, or bolded if ES is also $\geq 60$ or $\leq 40$. These cutoffs highlight datasets with the largest and most significant differences in $Z_C$ and $\bar{n}_{H_2O}$. Mean and median values of $Z_C$ and $\bar{n}_{H_2O}$ are given in file summary.csv in Dataset S1.

Counting the underlined and bolded MD values in Table 1, the number of datasets with a significant difference in $Z_C$ (18) is greater than those with a significant difference in $\bar{n}_{H_2O}$ (10). Of the 10 unique studies yielding at least one dataset with a significant difference in $Z_C$ in a comparison with normal tissue, 8 exhibit a higher mean value in adenoma or carcinoma compared to normal tissue. One of the other studies (*Besson et al.*, *2011*) has datasets with higher mean $Z_C$ in proteins up-expressed in adenoma, but lower mean $Z_C$ in proteins up-expressed in stage II and III carcinoma, compared to normal tissue. A second study, which analyzed proteins in stromal cells (*Li et al.*, *2016*), shows a significantly lower $Z_C$ in adenoma compared to normal tissue.

Most of the studies analyzed proteins in whole or microdissected tissue, but two datasets in studies from the same laboratory represent the nuclear matrix or chromatin-binding fraction (*Albrethsen et al.*, *2010*; *Knol et al.*, *2014*). These two datasets give lower mean $Z_C$ of proteins more highly expressed in carcinoma than adenoma. One other dataset has a lower mean $Z_C$ of proteins up-expressed in carcinoma compared to adenoma (*Wiśniewski et al.*, *2015*), and one has a higher mean value (*Mikula et al.*, *2011*).

The datasets with a significant difference in $\bar{n}_{H_2O}$ all show higher mean values for proteins in adenoma (5 datasets) or carcinoma (3 datasets) compared to normal tissue, up- expressed compared to down-expressed serum biomarker candidates (*Jimenez et al.*, *2010*), and secreted proteins of tumor tissue compared to normal tissue (*de Wit et al.*,
*2014*). Interestingly, none of the datasets with a significant difference in $\bar{n}_{H_2O}$ corresponds to a carcinoma/adenoma comparison.

Natural variability inherent in the heterogeneity of tumors, as well as differences in experimental design and technical analysis, may underlie the opposite trends in $Z_C$ between some datasets that compare the same stages of cancer. However, there is a preponderance of datasets with higher values of $Z_C$ and $\bar{n}_{H_2O}$ for the proteins up-expressed in adenoma or carcinoma compared to normal tissue.

## Compositional comparisons of microbial proteins

Summary data on microbial populations from four studies were selected for comparison here. First, in a study of 16S RNA of fecal microbiota, *Wang et al.* (*2012*) reported genera that are significantly increased or decreased in CRC compared to healthy patients. In order to compare the chemical compositions of the microbial populations, single species with sequenced genomes were chosen to represent each of these genera (see Table 2). Where possible, the species selected are those mentioned by *Wang et al.* (*2012*) as being significantly altered, or are species reported in other studies to be present in healthy or cancer states (see Table 2).

In the second study considered (*Zeller et al.*, *2014*), changes in the metagenomic abundance of fecal microbiota associated with CRC were analyzed for their potential as a biosignature for cancer detection. The species shown in Fig. 1A of *Zeller et al.* (*2014*) with a log odds ratio greater than 0.15 were selected for comparison, and are listed in Table 3. *Zeller et al.* (*2014*) found a strong enrichment of *Fusobacterium* in cancer, consistent with previous reports (*Kostic et al.*, *2012*; *Castellarin et al.*, *2012*). In a third study, *Candela et al.* (*2014*) reported the findings of a network analysis that identified 5 microbial "co-abundance groups" at the genus level. As before, single representative species were selected in this study, and are listed in Table 2. Except for the presence of *Fusobacterium*, the co-abundance groups show little genus-level overlap with community profiles derived from the previous two studies.

Finally, Table 4 lists the "best aligned strain" from Supplementary Dataset 5 of *Feng et al.* (*2015*) for all species shown there with negative enrichment in cancer, and for selected species with positive enrichment in cancer. Although every uniquely named strain given by *Feng et al.* (*2015*) was used in the comparisons below ($n = 44$; see Fig. 3D below), for clarity only the up-enriched species that appear in the calculated stability diagram (see Fig. 4D below) are listed in Table 4 and labeled in Fig. 3D. File `microbes.csv` in Dataset S1 contains the complete list of Bioproject IDs and calculated $Z_C$ and $\bar{n}_{H_2O}$ for all the microbial species considered here.

For each of the microbial species listed in Tables 2–4, a mean protein composition was calculated by combining amino acid sequences of all proteins downloaded from the NCBI genome page associated with the Bioproject IDs shown in the Tables (see file `microbial.aa.csv` in Dataset S1). This method does not account for actual protein abundances in organisms, and excludes any post-translational modifications. Calculation of the mean amino acid composition of proteins in this way is not an exact representation of the cellular protein composition, but provides a starting point for identifying environmental

**Table 2   Microbial species selected as models for genera and co-abundance groups that differ between CRC and healthy patients.**

| Phylum | Species | Abbrv. | Bioproject | Refs. |
|---|---|---|---|---|
| *Model species for genera significantly higher in healthy patients*[a] | | | | |
| Bacteroidetes | *Bacteroides vulgatus* ATCC 8482 | Bvu | PRJNA13378 | [c] |
| Bacteroidetes | *Bacteroides uniformis* ATCC 8492 | Bun | PRJNA18195 | [c] |
| Firmicutes | *Roseburia intestinalis* L1-82 (DSM 14610) | Rin | PRJNA30005 | [d] |
| Bacteroidetes | *Alistipes indistinctus* YIT 12060 | Ain | PRJNA46373 | [c] |
| Firmicutes | *Eubacterium rectale* ATCC 33656 | Ere | PRJNA29071 | [e] |
| Proteobacteria | *Parasutterella excrementihominis* YIT 11859 | Pex | PRJNA48497 | [f] |
| *Model species for genera significantly higher in CRC patients*[a] | | | | |
| Bacteroidetes | *Porphyromonas gingivalis* W83 | Pgi | PRJNA48 | [g] |
| Proteobacteria | *Escherichia coli* NC101 | Eco | PRJNA47121 | [c,h] |
| Firmicutes | *Enterococcus faecalis* V583 | Efa | PRJNA57669 | [c] |
| Firmicutes | *Streptococcus infantarius* ATCC BAA-102 | Sin | PRJNA20527 | [i] |
| Firmicutes | *Peptostreptococcus stomatis* DSM 17678 | Pst | PRJNA34073 | [j] |
| Bacteroidetes | *Bacteroides fragilis* YCH46 | Bfr | PRJNA58195 | [g] |
| *Model species for protective co-abundance groups*[b] | | | | |
| Actinobacteria | *Bifidobacterium longum* NCC2705 | Blo | PRJNA57939 | [g,k] |
| Firmicutes | *Faecalibacterium prausnitzii* SL3/3 | Fpr | PRJNA39151 | [e,l] |
| *Model species for pro-carcinogenic co-abundance groups*[b] | | | | |
| Fusobacteria | *Fusobacterium nucleatum* ATCC 23726 | Fnu | PRJNA49043 | [m,n] |
| Bacteroidetes | *Prevotella copri* DSM 18205 | Pco | PRJNA30025 | [k,o] |
| Firmicutes | *Coprobacillus* sp. D7 | Csp | PRJNA32495 | [h] |

**Notes.**

[a] Genus identification from Table 2 of *Wang et al.* (*2012*). Based on comments in *Wang et al.* (*2012*), *Bacteroides* is represented here by two species (*B. vulgatus* and *B. uniformis*) in healthy patients, and one species (*B. fragilis*) in CRC patients.

[b] Genus-level definition of co-abundance groups from *Candela et al.* (*2014*).

[c] *Wang et al.* (*2012*); species closely related to 16S rRNA-derived operational taxonomic units (OTUs; Fig. 2 of *Wang et al.*, *2012*) or otherwise mentioned by those authors (*E. faecalis*).

[d] *Duncan et al.* (*2002*).

[e] *Louis & Flint* (*2007*).

[f] *Nagai et al.* (*2009*).

[g] *Chen et al.* (*2012*).

[h] *Candela et al.* (*2014*).

[i] *Biarc et al.* (*2004*).

[j] *Zeller et al.* (*2014*).

[k] *Weir et al.* (*2013*).

[l] *Sokol et al.* (*2008*).

[m] *Castellarin et al.* (*2012*).

[n] *Kostic et al.* (*2012*).

[o] cf. *Chen et al.* (*2012*) and *Candela et al.* (*2014*) (more abundant in CRC patients); *Weir et al.* (*2013*) (more abundant in healthy subjects).

signals in protein composition. Mean amino acid compositions or amino acid frequencies deduced from microbial genomes, calculated without weighting for actual protein abundance, have been used in many studies making evolutionary and/or environmental comparisons (e.g., *Tekaia & Yeramian*, *2006*; *Zeldovich, Berezovsky & Shakhnovich*, *2007*; *Brbić et al.*, *2015*). In the future, more refined calculations may be possible by using genome-wide estimates of protein expression levels based on codon usage patterns (e.g., *Moura, Savageau & Alves*, *2013*; *Brbić et al.*, *2015*).

**Table 3  Species from a consensus microbial signature for CRC classification of fecal metagenomes** (*Zeller et al., 2014*). Only species reported as having a log odds ratio larger than ±0.15 are listed here, together with strains and Bioproject IDs used as models in the present study.

| Species | Strain | Abbrv. | Bioproject |
|---|---|---|---|
| **Higher in CRC patients** | | | |
| *Fusobacterium nucleatum* subsp. *vincentii* | ATCC 49256 | Fnv | PRJNA1419 |
| *Fusobacterium nucleatum* subsp. *animalis* | D11 | Fna | PRJNA32501 |
| *Peptostreptococcus stomatis* | DSM 17678 | Pst | PRJNA34073 |
| *Porphyromonas asaccharolytica* | DSM 20707 | Pas | PRJNA51745 |
| *Clostridium symbiosum* | ATCC 14940 | Csy | PRJNA18183 |
| *Clostridium hylemonae* | DSM 15053 | Chy | PRJNA30369 |
| *Lactobacillus salivarius* | ATCC 11741 | Lsa | PRJNA31503 |
| **Higher in healthy patients** | | | |
| *Clostridium scindens* | ATCC 35704 | Csc | PRJNA18175 |
| *Eubacterium eligens* | ATCC 27750 | Eel | PRJNA29073 |
| *Methanosphaera stadtmanae* | DSM 3091 | Mst | PRJNA15579 |
| *Phascolarctobacterium succinatutens* | YIT 12067 | Psu | PRJNA48505 |
| unclassified *Ruminococcus* sp. | ATCC 29149[a] | Rsp | PRJNA18179 |
| *Streptococcus salivarius* | SK126 | Ssa | PRJNA34091 |

Notes.
[a] *R. gnavus.*

**Table 4  Selected microbial species enriched or depleted in stool samples from cancer patients compared to healthy controls** (*Feng et al., 2015*).

| Enriched species | Abbrv. | Depleted species | Abbrv. |
|---|---|---|---|
| *Bacteroides dorei* | Bdo | *Actinomyces viscosus* | Avi |
| *Bacteroides ovatus* | Bov | *Bifidobacterium animalis* | Ban |
| butyrate-producing bacterium SS3/4 | But | *Clostridium* sp. SS2/1 | Csp |
| *Clostridium asparagiforme* | Cas | *Ruminococcus* sp. 5_1_39BFAA | Rsp |
| *Fusobacterium* sp. oral taxon 370 | Fsp | *Streptococcus mutans* | Smu |
| *Lachnospiraceae* bacterium 3_1_57FAA_CT1 | L57 | *Streptococcus thermophilus* | Sth |
| *Paraprevotella clara* | Pcl | | |
| *Peptostreptococcus stomatis* | Pst | | |
| *Ruminococcaceae* bacterium D16 | Rba | | |

The water demand per residue ($\overline{n}_{H_2O}$) vs. oxidation state of carbon ($Z_C$) in the mean amino acid compositions of proteins from all of the microbial species considered here are plotted in Figs. 1B and 1D, and for individual datasets in Fig. 3. The groups of proteins in the microbes enriched in cancer patients have somewhat lower $Z_C$ than those enriched in healthy donors in the same study. The dataset from *Feng et al.* (*2015*) (Fig. 3D) shows a more complex distribution, where the microbes with a relative enrichment in healthy individuals form two clusters at high and low $Z_C$. The *Fusobacterium* species identified in the studies of *Zeller et al.* (*2014*), *Candela et al.* (*2014*) and *Feng et al.* (*2015*) have the lowest $Z_C$ of any microbial species considered here. The mean human protein composition is also

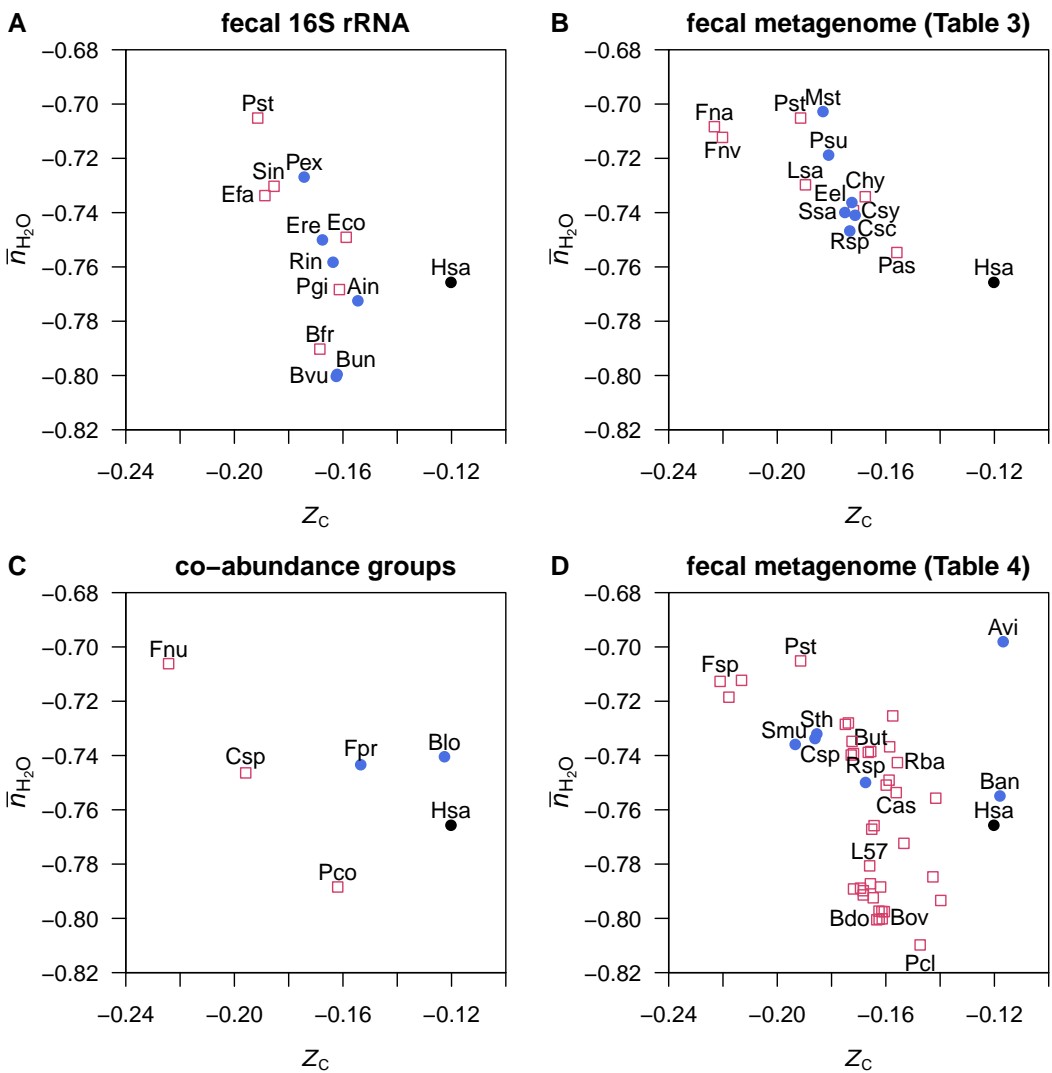

**Figure 3** Average oxidation state of carbon ($Z_C$) and water demand per residue ($\bar{n}_{H_2O}$) for mean amino acid compositions of proteins in genomes of normal- and cancer-enriched microbes. Data are shown for representative species for (A) microbial genera identified in fecal 16s RNA (*Wang et al., 2012*; Table 2 top), (B) microbial signatures in fecal metagenomes (*Zeller et al., 2014*; Table 3), (C) microbial co-abundance groups (*Candela et al., 2014*; Table 2 bottom), and (D) best aligned strains to metagenomic linkage groups in fecal samples (*Feng et al., 2015*; Table 4). The mean amino acid composition of proteins in the *Homo sapiens* genome (Hsa) is also shown.

plotted in Fig. 3, revealing a higher $Z_C$ than any of the mean microbial proteins except for *Actinomyces viscosus* and *Bifidobacterium animalis*, identified in the study of *Feng et al.* (*2015*) (Fig. 3D). The tendency for microbial organisms to be composed of more reduced biomolecules than the host may reflect the relatively reducing conditions in the gut.

## Thermodynamic descriptions: background

Going beyond compositional comparisons, thermodynamic descriptions can account for stoichiometric and energetic constraints and provide a richer interpretation of proteomic data in the context of tumor microenvironments.

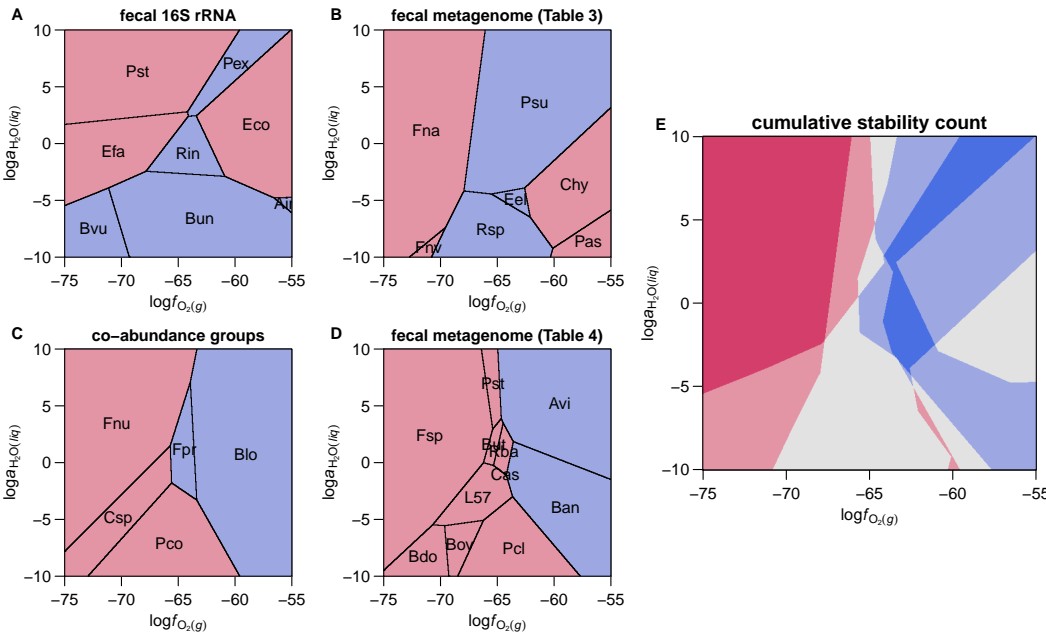

**Figure 4 Maximal relative stability diagrams for mean microbial protein compositions.** Each stability field in these diagrams shows the ranges of oxygen fugacity and water activity (in log units: $\log f_{O_2}$ and $\log a_{H_2O}$) where the mean protein composition from the labeled microbial species has a higher per-residue affinity (lower Gibbs energy) of formation than the others. Blue and red shading designate microbes relatively enriched in samples from healthy donors and cancer patients, respectively. Plot (E) is a composite figure in which the intensity of shading corresponds to the number of overlapping healthy- or cancer-enriched microbes in the preceding diagrams.

By combining both stoichiometric and energetic variables, a thermodynamic description of proteomic data reveals possible biochemical constraints that may arise within cells and in tumor microenvironments. To give an example of how relative stabilities of up- and down-expressed proteins in a proteomic dataset can be calculated as a function of chemical potentials, consider Reaction (R3) above written for the formation of one mole of MUC1. In order to compare proteins of different lengths, the formula of the protein is written per residue. The corresponding reaction is then

$$0.006C_3H_7NO_2S + 0.427C_5H_9NO_4 + 0.411C_5H_{10}N_2O_3$$
$$\rightarrow C_{4.203}H_{8.557}N_{1.253}O_{2.403}S_{0.006} + 0.714H_2O + 0.416O_2. \tag{R4}$$

An expression for the chemical affinity (*Kondepudi & Prigogine*, *1998*; *Helgeson et al.*, *2009*) of Reaction (R4) is

$$A = 2.303RT\log(K/Q) \tag{4}$$

where 2.303 is shorthand for the natural logarithm of 10, $R$ is the gas constant, $T$ is temperature, log represents the common (decimal) logarithm, and the activity quotient $Q$ is given by

$$\log Q = \log a_{C_{4.203}H_{8.557}N_{1.253}O_{2.403}S_{0.006}} + 0.714\log a_{H_2O} + 0.416\log f_{O_2}$$
$$- 0.006\log a_{C_3H_7NO_2S} - 0.427\log a_{C_5H_9NO_4} - 0.411\log a_{C_5H_{10}N_2O_3}. \tag{5}$$

The equilibrium constant is given by $-2.303RT\log K = \Delta G_r^o$, where $\Delta G_r^o$ is the standard Gibbs energy of the reaction. As noted above, the standard Gibbs energies of species used to calculate $\Delta G_r^o$ at $T = 37$ °C are generated using amino acid group additivity for the proteins and published values for standard thermodynamic properties of the basis species in the reaction.

To compare the potential for formation of metastable molecules, the per-residue formulas of the proteins are assigned equal activities (1). Then, Eqs. (4)–(5) show that the affinity, and hence relative potential for formation of different proteins, is a function of not only their amino acid composition (which determines the chemical formulas and standard Gibbs energies of the proteins used here), but also system parameters including temperature and the chemical potentials of the components. In this study, the chemical activities of the amino acid basis species are provisionally set to constant values ($10^{-4}$), while $\log f_{O_2}$ and $\log a_{H_2O}$ are considered to be adjustable parameters that are used as exploratory variables. The ranges of these variables shown on the diagrams are selected in order to encompass the stability boundaries between groups of proteins differentially enriched in cancer and normal samples.

There are combinations of chemical activities of basis species in Eq. (5) where the per-residue formation reactions of two proteins have an equal affinity, indicating equal chemical stability of the proteins. Other combinations of chemical activities of basis species give the result that one protein-residue formula has a higher affinity than the other(s), indicating greater stability of this protein. This concept provides an approach for constructing stability diagrams, which may be called the "maximum affinity method", that can be used to reproduce published equilibrium and metastable equilibrium diagrams for many inorganic and organic systems as shown by examples in the CHNOSZ package (*Dick*, *2008*) and is used below for microbial proteins. Because of the greater numbers of individual proteins in human proteomic datasets, a new method based on the difference in weighted sums of ranks of affinities is used here to compare the relative stabilities of groups of up- and down-expressed proteins in cancer.

### Relative stability fields for microbial proteins

Stability diagrams are shown in Figs. 4A–4D for the four sets of microbial proteins described above. The first diagram, representing significantly changed genera detected in fecal 16S rRNA (*Wang et al.*, *2012*; first part of Table 2), shows maximal stability fields for proteins from 5 species relatively enriched in healthy patients, and 3 species enriched in CRC patients. The other 4 proteins in the system are less stable than the others within the range of $\log f_{O_2}$ and $\log a_{H_2O}$ shown and do not appear on the diagram. The relative positions of the stability fields in Fig. 4A are roughly aligned with the values of $Z_C$ and $\overline{n}_{H_2O}$ of the proteins; note for example the high-$\log f_{O_2}$ positions of the fields for the relatively high-$Z_C$ *Escherichia coli* and *Alistipes indistinctus*, and the high-$\log a_{H_2O}$ position of the field for the high-$\overline{n}_{H_2O}$ *Peptostreptococcus stomatis*. Except for *E. coli*, the proteins from the species enriched in CRC occupy the lower $\log f_{O_2}$ (reducing) and higher $\log a_{H_2O}$ zones of this diagram.

In thermodynamic calculations for proteins from bacteria detected in fecal metagenomes (*Zeller et al.*, *2014*; Table 3), the mean protein compositions of 3 of 6 healthy-enriched

microbes and 4 of 7 cancer-enriched microbes exhibit maximal relative stability fields (Fig. 4B). Here, the cancer-associated proteins occupy the more reducing (*Fusobacterium nucleatum* subsp. vincentii and subsp. animalis) or more oxidizing (*Clostridium hylemonae*, *Porphyromonas asaccharolytica*) regions, while the proteins from bacteria more abundant in healthy individuals are relatively stable at moderate oxidation–reduction conditions.

For the bacterial species representing microbial co-abundance groups (*Candela et al.*, *2014*; second part of Table 2), all of the 5 mean protein compositions are present on the diagram (Figs. 4C). Here, the proteins from cancer-enriched bacteria are more stable at reducing conditions and those from healthy-enriched microbes are stabilized by oxidizing conditions.

A stability diagram for proteins of bacteria identified in a second metagenomic study (*Feng et al.*, *2015*) shows a similar result (Fig. 3D) for the 11 mean protein compositions with highest stability at some point the diagram. These patterns in relative stability again reflect the differences in $Z_C$ of the proteins, although in this case, a greater proportion of proteins (33 out of the 44 included in the calculations) are not found to have maximal stability fields. The resulting stability diagram is therefore a more limited portrayal of the available data.

Figure 4E is a composite representation of the calculations, in which higher cumulative counts of maximal stability of proteins from bacteria enriched in normal and cancer samples in the four studies are represented by deeper blue and red shading, respectively. According to this diagram, the chemical conditions predicted to be most favorable for the formation of proteins in many bacteria enriched in CRC are characterized by low $\log f_{O_2}$. Proteins from bacteria that are abundant in healthy patients tend to be stabilized by moderate values of $\log f_{O_2}$. Despite the differences in experimental design and microbial identification between studies, the thermodynamic calculations reveal a shared pattern of relative stabilities among the four datasets considered here.

## Relative stability fields for human proteins

Diagrams like those shown above that portray the maximally stable protein compositions are inadequate for analysis of larger datasets such as those generated in proteomic studies. It is apparent in Fig. 5 that only three different proteins up-expressed in cancer, from the 106 proteins in the KWA+14 dataset (chromatin-binding proteins in carcinoma/adenoma), are maximally stable across a range of $\log f_{O_2}$. However, visual inspection reveals a differential sensitivity to oxygen fugacity in the whole dataset, with lower $\log f_{O_2}$ providing relatively higher potential for the formation of many of the up-expressed proteins in carcinoma samples. How can these responses be quantified in order to explore the data in multiple dimensions, including both $\log a_{H_2O}$ and $\log f_{O_2}$?

In Fig. 5B, the difference in mean values of chemical affinity per residue of carcinoma and adenoma-associated proteins appears as a straight line as a function of $\log f_{O_2}$. This linear behavior would translate to evenly spaced isostability (taken as constant mean affinity difference) contours on a $\log f_{O_2}$–$\log a_{H_2O}$ diagram. The weighted rank difference of affinities (see Methods), shown by the curved line Fig. 5B, is a summary function that is more informative of changing chemical conditions. The variable slope is greatest near

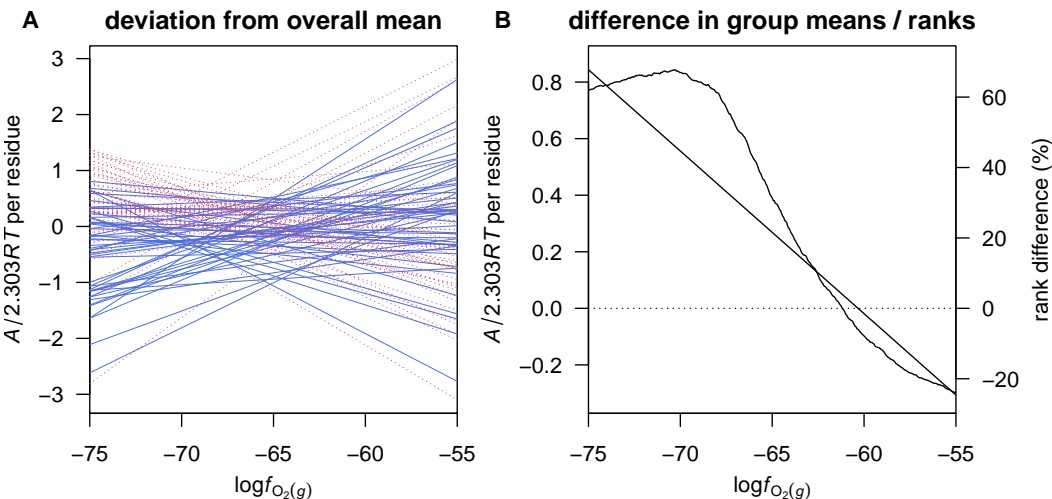

**Figure 5  Calculated chemical affinities per residue of proteins in the KWA+14 dataset.** Values for individual proteins as a function of $\log f_{O_2}$ at $\log a_{H_2O} = 0$ are shown in plot (A) as deviations from the mean value for all proteins. Down- and up-expressed proteins in carcinoma compared to adenoma are indicated by solid blue and dashed red lines, respectively. Plot (B) shows the difference in mean value between down- and up-expressed proteins (straight line and left-hand $y$-axis) and the weighted difference in sums of ranks of affinities as a percentage of maximum possible rank-sum difference (Eq. (3); curved line and right-side $y$-axis). Positive values of affinity or rank-sum difference in plot (B) correspond to relatively greater stability of the up-expressed proteins.

the zone of convergence for affinities of individual proteins (Fig. 5A), corresponding to the transition zone between groups of proteins. The resulting two-dimensional stability diagrams shown below have curved and diversely spaced isostability (taken as constant weighted rank difference of affinity) contours.

The diagrams in Fig. 6 portray weighted rank differences of chemical affinities of formation between groups of up- and down-expressed proteins reported for proteomic experiments. These combined depictions of stoichiometric and energetic differences constitute a theoretical prediction of the relative chemical (not conformational) stabilities of the proteins.

The slopes of the equal-stability lines and the positions of the stability fields reflect the magnitude and sign of differences in $Z_C$ and $\bar{n}_{H_2O}$. Figures 6A–6C show results for datasets that are dominated by differences in $\bar{n}_{H_2O}$; the nearly horizontal lines show that relative stabilities are accordingly more sensitive to $\log a_{H_2O}$ than $\log f_{O_2}$. The second row depicts relative stabilities in the three datasets from *Mikula et al.* (*2011*), which have large changes in, sequentially, $\bar{n}_{H_2O}$, $Z_C$, then both of these (Table 1). Accordingly, the equal-stability lines for these datasets are closer to horizontal, closer to vertical, or have a more diagonal trend (Figs. 6D–6F).

The last row shows results for datasets that are characterized by large changes in $Z_C$; the relative stabilities depend strongly on $\log f_{O_2}$. According to Fig. 6G, higher oxygen fugacity increases the relative potential for the formation of proteins up-expressed in cancer (dataset of *Jankova et al.*, *2011*). However, in a dataset for up- and down-expressed chromatin-binding proteins in carcinoma (*Knol et al.*, *2014*), lower $\log f_{O_2}$ is predicted to

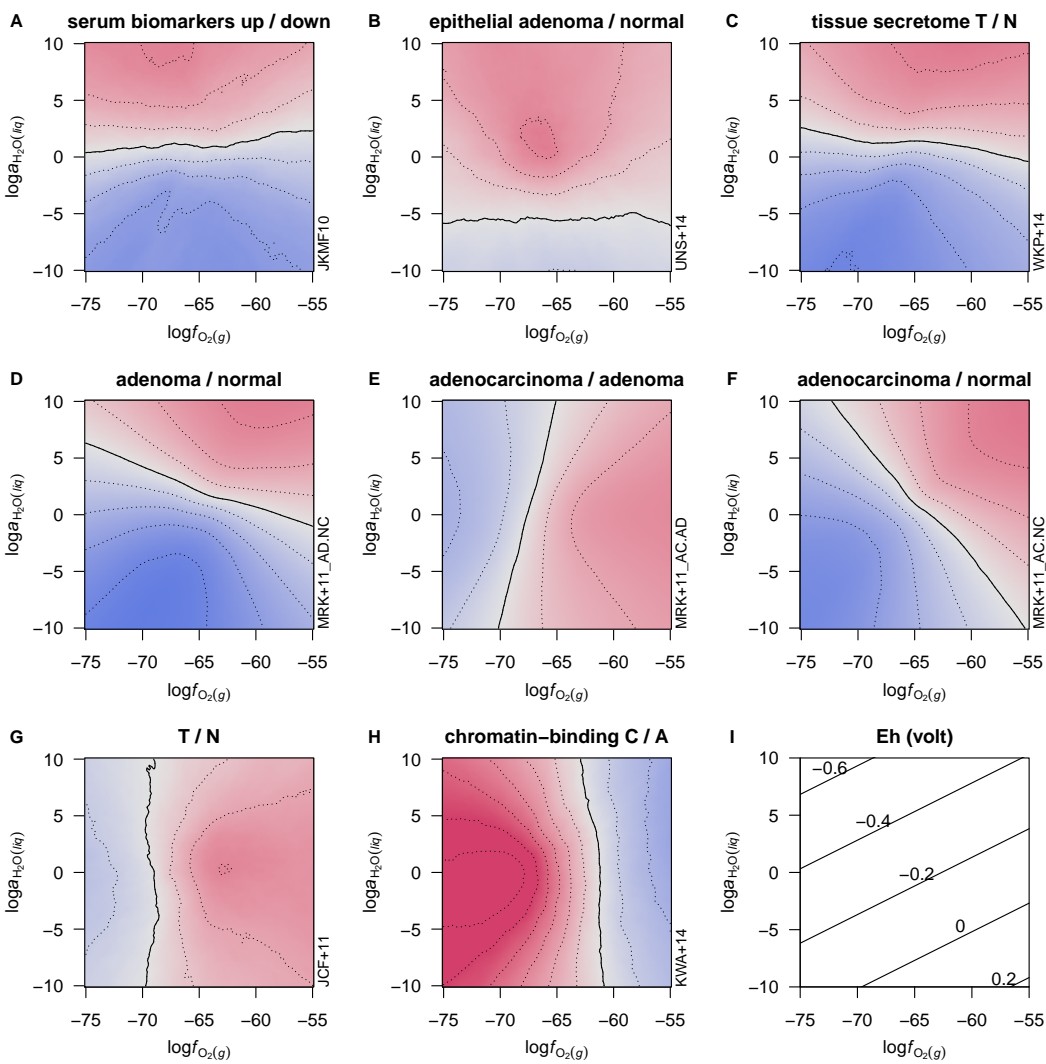

**Figure 6** **Weighted rank-sum comparisons of chemical affinities of formation of human proteins as a function of $\log f_{O_2}$ and $\log a_{H_2O}$.** The solid lines indicate equal ranking of proteins in the "normal" and "cancer" groups (Table 1), and dotted contours are drawn at 10% increments of the maximum possible rank-sum difference. Blue and red areas correspond to higher ranking of cancer- and normal-enriched proteins, respectively, with the intensity of the shading increasing up to 50% the maximum possible rank-sum difference. (For readers without a color copy: the stability fields for proteins up-expressed in cancer lie above (A–D), to the right of (E–G), or to the left of (H) the stability fields for proteins with higher expression in normal tissue.) Panel (I) shows calculated values of Eh over the same ranges of $\log f_{O_2}$ and $\log a_{H_2O}$ (cf. Reaction (R5)).

promote formation of the proteins up-expressed in carcinoma. This is the opposite trend to that found for most of the other datasets with significant differences in $Z_C$. These opposing trends might be attributed to different biochemical constraints acting at the subcellular and cellular or tissue levels during carcinogenesis.

The full set of diagrams for all datasets listed in Table 1 is provided in Fig. S1. It is notable that for the datasets where the relative stabilities are strongly a function of $\log a_{H_2O}$ (sub-horizontal lines), the equal-stability lines are within a few log units of 0 (unit activity).

Equal-stability lines that are diagonal often cross unit activity of $H_2O$ at a moderate value of $\log f_{O_2}$, near $-65$ to $-60$ (see Fig. S1). This could be indicative of a tendency for these proteomic transformations to be partially buffered by other redox reactions in the cell, and/or by liquid-like $H_2O$ with close to unit activity.

Effective values of oxidation–reduction potential (Eh) can be calculated by considering the water dissociation reaction, i.e.,

$$H_2O \rightleftharpoons \frac{1}{2}O_2 + 2H^+ + 2e^-. \tag{R5}$$

If one assumes that $\log a_{H_2O} = 0$ (unit water activity, as in an infinitely dilute solution), this reaction can be used to interconvert $\log f_{O_2}$, pH and pe (or, in conjunction with the Nernst equation, Eh) (e.g., *Garrels & Christ*, *1965*, p. 176; *Anderson*, *2005*, p. 363). However, in the approach utilized here for assessing the relative stabilities of proteins in a subcellular context, no such assumptions are made on the operational value of $\log a_{H_2O}$. Instead, it is used as an indicator of the internal state of the system, and is not necessarily buffered by an aqueous solution. Consequently, the effective Eh is considered to be a function of variable $\log f_{O_2}$ and $\log a_{H_2O}$, as shown in Fig. 6I for pH $= 7.4$ and $T = 37$ °C. This comparison gives some perspective on operationally reasonable ranges of $\log f_{O_2}$ and $\log a_{H_2O}$.

The subcellular reduction potential monitored by the reduced glutathione (GSH)/oxidized glutathione disulfide (GSSG) couple ranges from ca. $-260$ mV for proliferating cells to ca. $-170$ mV for apoptotic cells (*Schafer & Buettner*, *2001*), lying toward the middle part of the range of conditions shown in Fig. 6. A physiologically plausible Eh value of $-0.2$ V, corresponding to $\log f_{O_2} = -62.8$ at unit activity of $H_2O$, is close to the stability transitions for many of the datasets considered here (see also Fig. S1).

## Comparison with inorganic basis species

Figures made using Basis I (inorganic basis species, e.g., Reaction (R1)) are provided in the Supplemental Information (human proteins: Fig. S2; microbial proteins: Fig. S3). The stability boundaries in $\log a_{H_2O}$–$\log f_{O_2}$ diagrams constructed using Basis I cluster around a common, positive slope, in contrast with the greater diversity of slopes appearing on the corresponding diagrams constructed using Basis II (Fig. S1).

As noted above, all mathematically possible choices for the basis species of a system are thermodynamically valid, but it appears that Basis II affords a greater convenience for interpretation. That is, compared to Basis I, Basis II yields a greater degree of separation of the effects of changing chemical potentials of $H_2O$ and $O_2$ under the assumption that the activities of the remaining basis species (inorganic species in Basis I, or amino acids in Basis II) are held constant. However, it is also notable that two of the diagrams constructed using Basis I (Fig. S2), unlike the others, have nearly horizontal equal-stability lines, showing that increasing activity of $H_2O$ at constant activity of $CO_2$, $NH_3$, $H_2S$ and fugacity of $O_2$ gives an energetic advantage to the formation of potential up-expressed serum biomarkers (dataset JKMF10; *Jimenez et al.*, *2010*) and proteins up-expressed in an "epithelial cell signature" for adenoma (dataset UNS+14; *Uzozie et al.*, *2014*). These datasets are also found to be among those having significantly differential water demand using Basis II (Table 1; Fig. S1). Based

on the similar results for these datasets using different choices of chemical components, it can be suggested that the compositions of the differentially expressed proteins in these datasets are especially indicative of changes in hydration potential.

## DISCUSSION

Among 35 proteomic datasets considered here (Table 1), many have significantly higher values of average oxidation state of carbon ($Z_C$) in proteins up-expressed in adenoma or carcinoma compared to normal tissue. While a decrease in oxidation state might be expected if the differential expression of proteins was to some extent an adaptation to hypoxic conditions in tumors, the observed increase is more consistent with potentially oxidizing subcellular conditions that may accompany mitochondrial generation of ROS.

Available data for the adenoma to carcinoma transition are less conclusive: different datasets have relatively higher (*Mikula et al.*, *2011*) or lower (*Wiśniewski et al.*, *2015*) $Z_C$ of up-expressed proteins in carcinoma. A trend toward more reduced proteins in carcinoma compared to adenoma is also apparent in datasets for nuclear matrix fractions in chromosomal instability (CIN-type) CRC (*Albrethsen et al.*, *2010*) and for chromatin-binding fractions (*Knol et al.*, *2014*). It is possible that particular subtypes of cancer and/or subfractions of cells have patterns of protein expression during carcinogenesis that are chemically distinct from trends observed at the tissue level.

Some proteomic datasets are also available for stromal cells associated with tumor tissues. Data from one study (*Mu et al.*, *2013*) are consistent with the generally observed higher $Z_C$ of protein in tumors, but data from a pair of recent studies that analyzed cancer and stromal cells from the same set of tissues (*Li et al.*, *2016*; *Peng et al.*, *2016*) show that the proteins up-expressed in stromal cells, but not tumor cells, of adenoma are reduced compared to normal cells. Also, proteins up-expressed in tumor cells, but not stromal cells from carcinoma *in situ*, have a relatively oxidized composition (Table 1). If an opposing trend in $Z_C$ between stromal and epithelial cells is indeed established, it might be evidence for a proteome-level manifestation of metabolic coupling (*Martinez-Outschoorn, Lisanti & Sotgia*, *2014*) between tissue compartments in cancer. The "lactate shuttle" between metabolically coupled cells can be characterized in part by the difference between oxidation states of carbon in lactate ($Z_C = 0$) and pyruvate ($Z_C = 0.667$) (*Brooks*, *2009*). More work is needed to understand how the fluxes of anabolic precursors and catabolic products between tissue compartments might contribute to the differential oxidation states of carbon in proteins observed in cancer.

The datasets available for comparison of mean protein compositions of bacteria enriched in healthy subjects and cancer patients are characterized by lower $Z_C$ in proteins of bacteria with higher abundance in cancer patients (Fig. 3), and consequently stabilization of these proteins by lower oxygen fugacity ($\log f_{O_2}$; Fig. 4). This trend could be viewed as an adaptation of microbial communities to minimize the energetic costs of biomass synthesis in more reducing conditions. The opposite trends in $Z_C$ for the human and bacterial proteins also raises the possibility that their mutual proteomic makeup is partially the result of a redox balance, or coupling.

Another major outcome of the compositional comparisons of human proteomes is the increase in water demand per residue ($\bar{n}_{H_2O}$) apparent in some datasets for CRC tissues and in a list of candidate biomarkers summarized in a literature review (*Jimenez et al.*, *2010*) (Table 1). Higher hydration levels in breast cancer tissues have been observed spectroscopically (*Abramczyk et al.*, *2014*), and it has been proposed that increased hydration plays a role in reversion to an embryological mode of growth (*McIntyre*, *2006*). The thermodynamic calculations used to generate Fig. 6 support the possibility that higher water activity increases the potential for formation of the proteins up-expressed in cancer relative to normal tissue.

Although the ranges of $\log a_{H_2O}$ and $\log f_{O_2}$ derived from the model indicate to some extent the hydration and oxidation states of the system, they can not be interpreted directly in terms of measurable concentrations of water and oxygen. There are astronomical differences between theoretical values of oxygen fugacity in thermodynamic models and actual concentrations or partial pressures of oxygen (e.g., *Anderson*, *2005*, p. 364–365). Partial pressures of oxygen in human arterial blood are around 90–100 mmHg, and approximate threshold values for physiological hypoxia include 10 mmHg for energy metabolism, 0.5 mmHg for mitochondrial oxidative phosphorylation, and 0.02 mmHg for full oxidation of cytochromes (*Höckel & Vaupel*, *2001*). Assuming ideal mixing, the equivalent range of oxygen fugacities indicated by these measurements is $\log f_{O_2} = -4.57$ to $-0.88$, higher by far than the values that delimit the relative stabilities of cancer- and normal-enriched proteins computed here.

Likewise, the ranges of $\log a_{H_2O}$ calculated here deviate tremendously from laboratory-based determination of water activity or hydration levels. Water activity in saturated protein solutions is not lower than 0.5 (*Knezic, Zaccaro & Myerson*, *2004*), and recent experiments and extrapolations predict a range of ca. 0.600 to 0.650 for growth of various xerophilic and halophilic eukaryotes and prokaryotes (*Stevenson et al.*, *2015*). In general, cytoplasmic water activity is probably not greatly different from aqueous growth media, at 0.95 to 1 (*Cayley, Guttman & Record*, *2000*). The theoretically computed transitions in relative stabilities between proteins from cancer and healthy tissues occur at much lower values of $a_{H_2O}$ (ca. $10^{-6}$; Fig. 6B) or at values approaching 1, depending on the oxygen fugacity (Fig. 6; Fig. S1).

Despite the difficulties in a quantitative interpretation, theoretical predictions of stabilization of cancer-related proteins by an increase in $\log f_{O_2}$ (e.g., Figs. 6D–6G) can be interpreted qualitatively as corresponding with an increase in effective redox potential if $\log a_{H_2O}$ is held constant (Fig. 6I). Alternatively, proteins up-expressed in cancer tissues in each of the datasets shown in Figs. 6A–6G can be relatively stabilized along a trajectory of increasing both $\log f_{O_2}$ and $\log a_{H_2O}$ at constant effective redox potential near $-0.2$ V (Fig. 6I). Under this interpretation, local increases in both oxidation and hydration state are likely contributors to the proteomic transformations in colorectal cancer.

## CONCLUSION

An integrated picture of proteomic remodeling in cancer may benefit from accounting for the stoichiometric and energetic requirements of protein formation. This study has

identified a strong shift toward higher average oxidation state of carbon in proteins that are more highly expressed in colorectal cancer. This pattern is identified across multiple data sets, increasing confidence in its systematic nature. In some other data sets, a systematic change can be identified indicating greater water demand for formation of human proteins in cancer compared to normal tissue.

The proteomic data can be theoretically linked to microenvironmental conditions using thermodynamic models, which give estimates of the oxidation- and hydration-potential limits for relative stability of groups of proteins. These calculations outline a path connecting the dynamic compositions of proteomes to biochemical measurements such as Eh. This approach can be used in conjunction with other datasets to characterize chemical changes in proteomes in different types of cancer and in the progression to metastasis.

## ACKNOWLEDGEMENTS

Thanks to Greg Anderson for a helpful discussion about chemical components and Apar Prasad for giving many useful comments on the manuscript.

### Funding

The author received no funding for this work.

### Competing Interests

The author declares there are no competing interests.

### Author Contributions

• Jeffrey M. Dick analyzed the data and wrote the paper.

### Data Availability

The raw data has been supplied as Dataset S1.

### Supplemental Information

Supplemental information for this article can be found online at http://dx.doi.org/10.7717/peerj.2238#supplemental-information.

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
