# Peer review of "Proteomic indicators of oxidation and hydration state in colorectal cancer"

_PeerJ, doi:10.7717/peerj.2238_

## Round 0.1 · original submission · Major Revisions

· Academic Editor

Major Revisions

As you can see from reviewers' comments, there are significant concerns related to your analysis. For example, it was indicated that the description of the methods in the manuscript does not provide enough information on how the chemical formulae for proteins in cancer patients and healthy volunteers had been obtained. It was also pointed out that there is a substantial gap between published data and data required for calculations, and that this gap is not bridged in your study, leaving the reader wondering where all these data came from. Please note that although one of the reviewers recommended rejection, I decided to give you a chance to revise the manuscript and to answer critical points raised by both reviewers.

Reviewer 1 ·

Basic reporting

The language is clear and professionally used. Introduction is related to presented Methods and Results, Discussion sections. Figures are high quality and data are represented clearly. Literature is missing regarding hydration and redox studies on many proteins using statistical thermodynamics. These proteins have been investigated enormously using statistical thermodynamics without depending only on empirical formula. The structure of the manuscript is well designed based on which studies were shown from literature but a great part of literature is missing. Therewith, hydration studies using statistical mechanics are lacking not only in refereces but also in introduction, methods, results and discussions.

Experimental design

Experimental design depends on existing data from few groups without considering their weaknesses. It is a study that is directly linked to mistakes or weaknesses in existing data from few groups. For a scientist, the worst is to depend on existing studies and then applying empirical hydration calculations using simple mathematics. These proteins could be investigated independently from existing data in the literature shown by the author. One needs to employ statistical thermodynamics with dynamics in water for such purposes. The Zc numbers can be then predicted without depending on initial data from other groups. Therefore, I suggest a statistical mechnaics and then quantum mechanics studies for more advanced Zc numbers. The quantum calculations can be performed on smaller moities from molecular dynamics simulations without depending on existing data. Existing methods are too weak in their current forms. These methods also open the door for pandora given that statistics affected also previous studies, which were taken as data in this work to produce some related and relative data without dynamics and solvent effects with dynamics.

Validity of the findings

The findings are valid when we accept the methods. However, I have huge issues in accepting the methods part as chemical thermodynamics.

Comments for the author

Please revise your studies using statistical mechanics and quantum mechanics and then compare to existing data rather than using the literature as input data. Chemical Thermodynamics is more complicated.

Reviewer 2 ·

Basic reporting

The main hypothesis of the present work states that the local hypoxia and the hydration level in a tumor may affect the gene expression and protein composition of the cancerous cells. In order to characterize local parameters the author used indirect unconventional method of calculation the average carbon oxidation level and the water content per residue based exclusively on the chemical formulae of proteins synthesized in cancerous cells or adjacent bacteria.

This method had been originally developed as a tool in Geobiology to characterize the average carbon oxidation level in remnants of organic matter in the lithosphere. The author previously showed that this approach can be expanded to the field of biology and medicine. (Dick, J. M. (2014). Average oxidation state of carbon in proteins. Journal of the Royal Society Interface, 11:20131095. DOI 10.1098/rsif.2013.1095.) However, in that paper the author did not prove that this expression can be equally applied to large proteins found in living cells. There are only two self-references on this paper found by Google Scholar that indicates that this method of calculation of the average carbon oxidation in proteins had not been accepted and independently verified by other investigators. It is highly questionable that the approach, which is valid for simple organic molecules, can be applied to large complex molecules of proteins in the living cell.

The paper is well-written, although biologists and biochemists may find the language to be too complex. Introduction gives comprehensive background and provides detailed considerations of the problem in question. Throughout the manuscript the author gives proper citations relevant to the study. Figures are of good quality and correctly illustrate the main results obtained by the author.

Experimental design

This study is a pure theoretical one and relies exclusively on experimental data that the author found in the published literature. However, the description of the methods in the manuscript does not provide enough information on how the chemical formulae for proteins in cancer patients and healthy volunteers had been obtained. For example, the data published on the composition of colon bacteria in health and disease list only the relative change of bacteria genera present in the colon. There is no information on the particular proteins expressed in each bacterial species, thus there is no way to know their chemical formulae, which are required for calculations.

The use of overall genome data for bacteria present in the colon “without any weighting for transcript or protein abundance in organisms, and excluding any post-translational modifications” indicates that the author could not use formulae for these proteins to calculate the average carbon oxidation according to his expression (1).

The manuscript does not contain any description of the methods used to generate Figures 4 and 5. Namely, the source of oxygen fugacity (fO2) and water activity (aH2O) used to construct diagrams on Figure 4 is not indicated in the manuscript. This omission makes the section on the thermodynamic stability of bacterial proteins and the main conclusion of the manuscript rather questionable.

Validity of the findings

No Comments

---

## Round 0.2 · accepted · Accept

· Academic Editor

Accept

Special thanks for addressing all critical points raised by both reviewers.